# Potential of Synthetic and Natural Compounds as Novel Histone Deacetylase Inhibitors for the Treatment of Hematological Malignancies

**DOI:** 10.3390/cancers15102808

**Published:** 2023-05-17

**Authors:** Dilipkumar Pal, Khushboo Raj, Shyam Sundar Nandi, Surajit Sinha, Abhishek Mishra, Arijit Mondal, Ricardo Lagoa, Jack T. Burcher, Anupam Bishayee

**Affiliations:** 1Department of Pharmaceutical Sciences, Guru Ghasidas Vishwavidyalaya (A Central University), Bilaspur 495 009, India; 2Department of Biotechnology, Indian Council for Medical Research-National Institute of Virology, Mumbai 400 012, India; 3Department of Cancer Biology, Memorial Sloan Kettering Cancer Center, New York, NY 10065, USA; 4Aurobindo Pharma, Durham, NC 27703, USA; 5Department of Pharmaceutical Chemistry, M.R. College of Pharmaceutical Sciences and Research, Balisha 743 234, India; 6Associate Laboratory in Chemical Engineering, Polytechnic Institute of Leiria, Morro do Lena, Alto do Vieiro, 2411-901 Leiria, Portugal; 7College of Osteopathic Medicine, Lake Erie College of Osteopathic Medicine, Bradenton, FL 34211, USA

**Keywords:** leukemia, multiple myeloma, DNA damage, HDAC inhibitors, oxidative stress, chromatin remodeling, natural agents

## Abstract

**Simple Summary:**

Histone deacetylases (HDACs) are epigenetic regulators that influence chromatin structure and gene transcription, but also the function of non-histone targets like chaperones and proteins participating in the DNA damage response. Histone modifications are implicated in cancer and HDAC inhibitors (HDACis) can be useful to treat hematological malignancies, including leukemia, B-cell lymphoma, virus-associated tumors, and multiple myeloma. Several HDACis were approved for clinical use and others are showing promising results, especially as adjuvants to conventional treatments. In addition to synthetic inhibitors, this work discusses the potential of phytocompounds, such as chrysin and oleacein as well as marine-derived agents, such as halenaquinone and xestoquinone, as therapeutic agents against hematological cancers.

**Abstract:**

Histone deacetylases (HDACs) and histone acetyltransferases (HATs) are enzymes that remove or add acetyl groups to lysine residues of histones, respectively. Histone deacetylation causes DNA to more snugly encircle histones and decreases gene expression, whereas acetylation has the opposite effect. Through these small alterations in chemical structure, HATs and HDACs regulate DNA expression. Recent research indicates histone deacetylase inhibitors (HDACis) may be used to treat malignancies, including leukemia, B-cell lymphoma, virus-associated tumors, and multiple myeloma. These data suggest that HDACis may boost the production of immune-related molecules, resulting in the growth of CD8-positive T-cells and the recognition of nonreactive tumor cells by the immune system, thereby diminishing tumor immunity. The argument for employing epigenetic drugs in the treatment of acute myeloid leukemia (AML) patients is supported by evidence that both epigenetic changes and mutations in the epigenetic machinery contribute to AML etiology. Although hypomethylating drugs have been licensed for use in AML, additional epigenetic inhibitors, such as HDACis, are now being tested in humans. Preclinical studies evaluating the efficacy of HDACis against AML have shown the ability of specific agents, such as anobinostat, vorinostat, and tricostatin A, to induce growth arrest, apoptosis, autophagy and cell death. However, these inhibitors do not seem to be successful as monotherapies, but instead achieve results when used in conjunction with other medications. In this article, we discuss the mounting evidence that HDACis promote extensive histone acetylation, as well as substantial increases in reactive oxygen species and DNA damage in hematological malignant cells. We also evaluate the potential of various natural product-based HDACis as therapeutic agents to combat hematological malignancies.

## 1. Introduction

Approximately 1% of the DNA sequences within the human genome are responsible for coding the 20,000–25,000 genes of functional protein products. Noncoding DNA serves as binding sites for transcription factors, which contributes to a transcriptome that is many times larger than the number of genes. Still, further diversification of the proteome occurs by post-translational modifications (PTMs) of amino acid side chains. Most PTMs are considered to be dynamic in nature, meaning that an individual protein may be found in one of many different states and have the freedom to flow through each of them [1]. The α-amino side chain of lysine, for example, is the focus of many PTMs through alkylation or acylation processes. In contrast to acylation, which adds carbon to proteins and results in neutral or negatively charged side chains [2], alkylation augments the lysine side chain while maintaining its positive charge within a normal pH. Two PTMs, methylation and acetylation, occur in histone proteins, which contain a lysine-rich N-terminal, and play a major role in chromatin remodeling and management of binding partners to influence transcription [3,4]. For this reason, modifiers of lysine acetylation enzymes and agents promoting their elimination have become prime candidates for small molecule therapeutic development. 

Lysine acetylation was originally identified in the early 1960s when scientists noticed large amounts of PTMs in histones [5]. Soon afterwards, enzymes that catalyze the forward acetylation process and the reverse deacetylation reaction were found and named histone acetyltransferases (HATs) and histone deacetylases (HDACs), respectively [5]. As epigenetic regulators, HDACs control histone tail length, chromatin structure, protein-DNA interactions, and gene transcription. Additionally, proteins other than histones are the targets for HATs and HDACs. These include chaperone proteins, transcription factors, hormone receptors, signaling mediators, and other proteins implicated in the DNA damage response [6]. Given their extensive role in genetic regulation, it is unsurprising that histone modifications are implicated in numerous pathophysiologic states [6]. Many of these changes are observed in hematological malignancies and may be remedied, in part, by HDACis [7]. For example, the following changes are observed in lymphomas and leukemias: Cancer cells express changes in the global pattern of acetylationLevels of HDACs are increased in lymphoma cellsHDACs are abnormally attracted to the promoter of a target gene, where they suppress transcription and prevent differentiation, thereby contributing to the generation of acute promyelocytic leukemia.

Epigenetics refers to a multitude of changes in gene expression not involving modifications of the underlying DNA sequence [8,9]. As mentioned previously, lysine acetylation is a key histone chemical change necessary for epigenetic gene expression. HATs and HDACs are targets for cancer treatment because of their roles in reversibly regulating histone acetylation. HDACs’ inhibitors (HDACis) are a large and varied family of compounds found throughout nature and as synthetic products. Many different agents are being studied for their potential as HDACis; n-butyrate was the first to be recognized as the agent responsible for the accumulation of hyper-acetylated histone within the nucleus. Later, it was shown that HDACs might be inhibited in two more ways: irreversibly by trapoxin A and reversibly by trichostatin A (TSA). Inhibition of HDAC activity is the biochemical mechanism through which HDACis raise histone acetylation levels [10]. Several proteins associated with cancer development, apoptosis, cell cycle regulation, angiogenesis, and cell invasion are all targets of HDACs. HDACis have been demonstrated to support cell cycle arrest and induce apoptosis by triggering both the intrinsic and the extrinsic apoptotic pathways. HDACis may cause differentiation and/or death by blocking the cell cycle during the G1/S or G2/M transition. Moreover, their increased expression of p21WAF1/CIP1 (a cyclin-dependent kinase (CDK) inhibitor) causes cell cycle arrest at the G1/S phase [11]. 

The primary focus of this review is on the role of HDACis, including those derived naturally from plants, in the treatment of hematological cancers. Hematological cancers, also known as blood cancers, are a diverse group of malignancies that affect the production and functions of blood cells [12]. These cancers include leukemia, lymphoma, multiple myeloma, and other related conditions. HDACis, a form of chemotherapy, have been used in the treatment of hematological cancers. While they have shown promise in improving survival rates, they can also cause significant side effects [13].

One common side effect of HDAC inhibitors in the treatment of hematological cancers is myelosuppression, which is a reduction in the production of blood cells. This can lead to anemia, neutropenia, and thrombocytopenia, which can increase the risk of infections, bleeding, and other complications. Other side effects include fatigue, gastrointestinal symptoms, cardiac toxicity, and neuropathy [13].

As an alternative, natural compounds have been explored as potential therapies for hematological cancers. Some of these compounds have been shown to possess anticancer properties without the toxic side effects associated with traditional agents. For example, curcumin, which is found in turmeric, has been shown to induce apoptosis (cell death) in leukemia and lymphoma cells. Resveratrol, another phytochemical that is found in grapes, has been shown to inhibit the growth of multiple myeloma cells [14].

Other natural compounds that have shown promise in preclinical studies for the treatment of hematological cancers include green tea polyphenols, gingerol, quercetin, and luteolin. While more research is needed to determine the effectiveness of these compounds in humans, they offer a potential avenue for the development of safer and more effective treatments for hematological cancers [15]. 

It has also been hypothesized that dysregulated epigenetic enzymes and the resulting aberrant epigenetic modifications are intimately connected to tumor development and progression. The comprehensive mechanisms of action, categorization, and association of HDACs with various diseases, such as prostate, breast, and ovarian cancer have been previously studied [12,16]. In breast cancer cells, histone modifications may aid in maintaining genome integrity, DNA repair, transcription, and chromatin modulation [17]. After noticing their presence in a variety of medicines, Hieu et al. [18] created quinazolin-4(3H)-one-based HDACis. Sixteen synthetic compounds were evaluated against HDACs and cancer cells. The most potent HDACis had a half maximal inhibitory concentration (IC_50_) value of 90 nM, whereas the same molecule had an IC_50_ value of 810 nM against human prostate cancer PC-3 cells. Twelve agents containing hydroxyacetamide and triazole moieties were evaluated against HDACs and MCF-7 cells in a study published by Saha et al. [19]. Some of the designer drugs were hybrids of coumarin and vorinostat and were found to inhibit the expression of the tumor suppressor p53 and increase p21 in prostate and breast cancer models [20]. 

Epigallocatechin gallate (EGCG), a natural flavonoid found in green tea, has powerful anticancer activities through epigenetic processes, including histone modification [21]. In human colon cancer cell lines, 48 and 72 h treatment with 50 to 150 μM of EGCG was shown to decrease the levels of HDAC1-3 [22]. Notably, HDAC1 expression was less responsive, but EGCG downregulated protein levels of HDAC2 and 3 by more than 50% at a concentration of 100 μM [22]. EGCG (40 μM) and pEGCG (20 μM), a pro-drug of EGCG, significantly suppressed the growth of MDA-MB 231 and MCF-7 breast cancer cells [23]. 

Apigenin (4′,5,7-trihydroxyflavone) is another flavonoid derived from various natural sources. In PC-3 cells, the flavone was found to function as an HDACi at concentrations of 20–40 μM. Researchers found the treatment of cell lines to result in 41% and 62% suppression of HDAC activity, with the most profound effects seen in HDAC1 and HDAC3. Furthermore, apigenin promoted acetylation of histone H3 (more than 7-fold) and, less markedly, of H4 globally, while also centering the hyperacetylation of H3 in the p21 WAF1 promoter [24].

Although the impact of epigenetic modification in hematologic malignancies, including leukemia, lymphoma, and multiple myeloma, has been shown, few studies have fully examined the role of HDACis in this context [25,26]. Histone acetylation is an essential mediator of hematopoiesis that is closely tied to malignancy. This review aims to describe the physiologic effects of HDACis and their regulatory role in the development of blood cancers.

## 2. Classification of HDACs and Inhibitors

The 18 HDACs found in humans are organized into four classes based on their cellular locations, enzyme activity, and similarity to HDACs present in yeast (Table 1). Based on their chemical structures and enzymatic activity, HDACis are often divided into five categories: benzamides, aliphatic acids, cyclic tetrapeptides, hydroximates, and electrophilic ketones. HDACis may target one or more particular HDAC isoforms (HDAC isoform selective inhibitors) or all HDAC isoforms (pan-inhibitors). 

Zinc-dependent HDACis have a structure comprised of three domains: (i) a surface recognition unit, or “cap group”, (ii) a zinc-binding domain, (iii) a linker domain which connects the previous domains. HDACi selectivity is determined by the cap and linker domains, which participate in the ligand-enzyme interactions; the zinc-binding domain inhibits HDACis activity by binding to the zinc ion. The majority of human HDACs are zinc-dependent metalloenzymes, known individually as HDAC1 through HDAC11. These enzymes use water as a nucleophile to hydrolyze the amide bond [27]. The acyl group is transferred to the C2 position of the ribose sugar by the remaining seven HDACs, referred to as sirtuins (individually as sirtuins 1–7), which use NAD+ as a cofactor. Although members of both enzyme families carry out the identical chemical process of acyllysine cleavage, this discussion will be focused on zinc-dependent HDACs present in classes I, II, and IV (Table 1). HDACis may be categorized into different types based on their chemical structures and cancer specificity (Table 2).

## 3. Induction of DNA Damage by HDAC Inhibitors

Bakkenist et al. [40] suggested that the DNA damage response (DDR) can be activated independently of DNA double-strand breaks (DSBs). Their work showed that the activation of ataxia-telangiectasia mutated (ATM) kinase, the first step in the cellular response to ionizing radiation (IR), did not need direct binding to DNA DSB but rather was the consequence of alterations in chromatin structure. Other studies have shown evidence that HDACis may generate significant DNA damage, including DSBs, by increasing reactive oxygen species and oxidative stress and/or variant histone H2AX phosphorylation of serine, gH2AX, which is associated with DSBs (Figure 1) [41]. New studies have also demonstrated that modifications to chromatin may directly or indirectly result in DNA damage. This suggests the mechanisms responsible for chromatin remodeling and DNA damage signaling are intrinsically related [42]. 

Researchers have demonstrated that the application of HDACis to leukemic cells led to the acetylation of histone and prompt production of gH2AX [43]. After 24 h, the acute myeloid leukemia (AML) cell line HL-60 and healthy peripheral blood lymphocytes stimulated with IL-2 (PBL) were exposed to different concentrations of apicidin or TSA. Phosphorylation of H2AX emerged at 3 min, peaked at 30 min, and remained evident at 24 h following TSA (300 nM) treatment in HL-60 cells, but disappeared several hours following treatment in normal PBL. During this transition, acetylation of histone H4 increased on chromatin fibers in both HL-60 and PBL. Myeloid leukemia cell lines K562 and NB4 and primary cells from individuals with AML showed identical outcomes, highlighting the significance of these findings [44,45,46]. 

Investigators also found that TSA exposure to HL-60 cells led to enhanced phosphorylation of ATM. After 10 min, phosphorylation was at its highest and gradually decreased over the next 8 h. It is now well-established that induction of apoptosis is a primary mechanism of HDACis anticancer actions in leukemic cells, and all of the HDACis have been found to trigger apoptotic processes [47,48,49,50,51,52,53]. It was also observed that, by enhancing histone acetylation, activation of gH2AX and ATM phosphorylation occurred long after the initiation of apoptosis, supported by activation of caspase-3 and also poly(ADP-ribose)polymerase1 (PARP1) in HL-60 cells as early as 4 h after HDACis treatment. However, HDACis still produced early and long-lasting DNA damage, evidenced by gH2AX, when HL-60 and NB4 (AML stage M2 and M3) cells were pretreated with a pan-caspase inhibitor, which effectively blocked endogenous caspase activity [54]. Therefore, it seems that HDACis-dependent induction of DNA damage does not follow apoptosis but rather it occurs before. Moreover, trypan blue exclusion assays revealed that cell death was suppressed in cells pretreated with the caspase inhibitor and then treated with HDACis [55]. All of these studies help to conclude that HDACis selectively induce DNA damage and activate the DDR in neoplastic cells, which in turn leads to apoptotic processes [56]. 

When compared to other HDACis, such as sodium butyrate or trichostatin A, depsipeptide (FR901228 or FK228) is a stronger inhibitor of class I HDACs. This makes it a potential anticancer drug for both mono- and polytherapy. Depsipeptide exhibits additional biological effects through a variety of modes of action, beyond promoting histone acetylation. Non-histone proteins HSP90 and p53 are acetylated when exposed to depsipeptide. Nucleoside analogues, such as cytarabine and 5-aza-2-deoxycytidine (or 5-aza-CdR), work in tandem with depsipeptide to stimulate transcription and cause apoptosis [57,58,59]. 

Apoptosis presumably results from limited repair of DNA damage caused by 5-aza-CdR. In addition, depsipeptide has been shown to be a DNA demethylating agent, an observation that has broadened the scope of its research. The method by which depsipeptide kills human cancer cells may be deduced from these data. Almost all HDACis cause a dramatic upregulation of p21, a protein that causes G1 cell cycle arrest when sufficiently accumulated. It was previously observed that depsipeptide stimulates p21 expression through the elaboration of p53 pathways. Specifically, p53 acetylation at K373/382 may induce p53 trans-activity and expression of p21 in cells treated with depsipeptide. Other HDACis, including TSA, can only promote p53 acetylation in conjunction with nicotinamide (a class III HDACis) or ionizing radiation [60,61,62].

## 4. DNA Double-Strand Breaks

In mammalian cells, DSBs are a very harmful form of DNA damage. Due to the loss of duplex integrity, DSBs are more difficult to repair than other kinds of DNA damage and are often lethal for the cell. Ataxia-telangiectasia, a cancer-prone, radiation-sensitive human illness, is caused by pathogenic ATM variants, a protein pivotal role in the activation of cell cycle checkpoints following DSBs. Important for bringing ATM to DSBs, the complex of MRE11-RAD50-NBS1 (MRN) causes ATM to autophosphorylate, changing it from a dimer to a functional monomer. After activation, ATM phosphorylates MRN and subsequent effector proteins to prompt G1/S, intra-S and G2/M cell cycle checkpoints [63]. The activation of these checkpoints improves cell survival and genomic integrity, because it offers more time for DNA damage repair before it may be duplicated or passed on to daughter cells. Emerging data suggests that ATM not only participates in these DSB pathways, but also facilitates repair of DSBs in G0 and G2 cells. There are two major types of mechanisms for repairing DSBs, known as homology-directed repair (HRR) and nonhomologous end-joining (NHEJ) [64]. 

### 4.1. Homology-Directed Repair

HRR describes the repair of DSBs associated with replication. Occurring in the S and G2 phases, this repair process uses a template of identical DNA segments obtained from the sister chromatid, a repair pathway often error-free. The first step of HRR is the resection of the DSB using the human MRN complex and CtIP. Then, various accessory proteins, including BRCA2, hRad52, XRCC2, and XRCC3, help recruit hRad51 and assemble it into a nucleoprotein filament. Ultimately, two identical sister chromatids are produced when hRad51 nucleoprotein filaments are recruited and assembled [65].

### 4.2. Nonhomologous End-Joining

The NHEJ mechanism of DSBs repair begins with a ring-shaped protein complex called the Ku70/Ku86 heterodimer. This complex shields the damaged DNA ends from degradation by binding tightly to them. Following recognition of damaged DNA by the Ku heterodimer, DNA PK and other proteins are recruited. DNA PK undergoes a conformational change to produce active DNA protein kinase (PK), which recruits DNA ligase IV/XRCC4. This ligase completes the repair by connecting the DNA ends. Microhomologies, small complementary sequences present at the break point, are thought to aid in aligning the ends during DNA PK-dependent NHEJ. Larger deletions and chromosomal translocations may be produced by using an alternative form of NHEJ. When compared to DNA PK-mediated NHEJ, the alternative route is characterized by greater deletions and insertions, augmented microhomology regions and more chromosomal translocations. A variety of proteins involved in alternative NHEJ (PARP1, complex MRN, DNA ligase IIIa/XRCC1) are upregulated in malignant cells, though their effects in this context are poorly characterized [66].

## 5. Downregulation of DSB Repair

The implications of HDACis are greater than that of histone modification alone. Vorinostat (or SAHA) has been demonstrated to upregulate and maintain gH2AX expression in prostate and lung cancer cells by reducing the expression of MRE11 and HRR factor RAD50. In addition to RAD50, vorinostat was reported to reduce the NHEJ proteins Ku70 and Ku80 in melanoma, while NaB has been shown to downregulate Ku80, Ku70, and DNA PK [67]. Additional genes downregulated in AML cells by HDACis MS275 and LAQ824 include RAD50, BRCA1, CHK2, EXO1, and Ku80. Researchers started by looking at the genes involved in DNA repair and found that RAD51, BLM, CHK1, and BRCA1 were all downregulated in prostate cancer cells following SAHA and valproic acid treatment. Immunofluorescence analysis of DSB repair efficiency indicated that not only the expression of RAD51 and BRCA1 decreased, but also the correct targeting of the DNA repairing points was affected. In accordance, it was observed higher DNA damage, determined by the comet test and the development of gH2AX sites, and a reduction in DSB repair efficiency. E2F transcription factor 1 (E2F1), an established activator of CHK1, RAD51 and BRCA1 gene expression, was also found to be suppressed by HDACis. Researchers utilized chromatin immunoprecipitation experiments to verify that E2F1 recruitment to the promoters of CHK1, RAD51, and BRCA1 was compromised in the presence of HDACis [68].

## 6. Phenomenon of Chromatin Remodeling

Chromatin remodeling has been implicated in DNA damage repair by many independent trials. After DSBs are induced, gH2AX is activated and localized to the damaged sites. gH2AX stimulates the recruitment of MDC1 to DNA DSBs, which attaches ATM to the chromatin by means of NBS1 activation. The MRN complex aids in preparing the damaged sites to repair, whereas CK2 phosphorylation of MDC1 coordinates subsequent modifications to chromatin. When gH2AX is modified, MDC1 triggers the recruitment of the NuA4 complex, which includes the HIV1 HAT Tat interacting protein 60 kDa (TIP60-p400). TIP60, a HAT, acetylates histones H2A and H4K16, which loosens the chromatin and makes it easier for DNA repair proteins to reach sites of DNA damage [69]. 

Sharma et al. [70] demonstrated that the development of IR-induced gH2Ax is reliant on MOF activity, but its removal is under the control of TIP60. Application of HAT inhibitor anacardic acid to IR-treated lung cancer cells resulted in a reduced recruitment of Ku70 and Ku80 NHEJ proteins, presumably due to silencing or suppression of P300/CBP. The significance of chromatin relaxation in the DNA repair response is further emphasized by the fact that the SWI/SNF chromatin remodeling complex, which is known to enhance NHEJ through the local relaxing of chromatin, is not recruited after treatment with a HAT inhibitor [71].

## 7. Hematological Cancers

Hematological cancers are those that affect the blood, bone marrow, and lymph nodes. Depending on the kind of cell of origin, they are designated as leukemia, lymphoma, or myeloma (Figure 2). The immune system interconnects these three forms of cancer, and a sickness that affects one often impacts the others [72].

## 8. Role of HDAC Inhibitors in Hematological Cancers

### 8.1. Acute Myeloid Leukemia

AML is a clonal illness defined by the accumulation of poorly differentiated (immature) myeloid cells in the peripheral blood and bone marrow due to uncontrolled replication. AML is a type of adult acute leukemia, the most prevalent, showing a median age of onset of 70 years. Although survival metrics have increased in recent years, additional pharmacological options are needed to improve long-term survival rates. Survival is substantially worse in older patients, which is compounded by AML’s late age of onset [73]. 

Despite substantial progress in our understanding of the genetics surrounding its pathogenesis, treatment of AML has remained relatively consistent for the last four decades and consists of conventional chemotherapy and allogeneic hematopoietic stem cell transplantation. In recent years, the United States Food and Drug Administration (FDA) has approved novel medications for AML, including ivosidenib, ozogamicin, gemtuzumab, venetoclax, enasidenib, glasdegib, midostaurin, and glitertinib, summarized in Table 3.

Historically, AML was believed to be a hereditary illness; however, a number of studies have been reinforcing the contribution of epigenetic alterations in the disease. DNA methylation and histone modification are well-studied epigenetic changes linked with chromatin structure and gene expression. Epigenetic regulatory enzymes responsible for DNA and histone modifications are implicated in the pathogenesis of hematological malignancies. In fact, most of these enzymes have been proven to be unregulated or otherwise altered in many malignancies, especially in AML [74]. Encouraged by the findings that epigenetic alterations can be reversed, epigenetic medicines have attracted great interest in the development of AML therapies [75]. HDACis promote histone acetylation, which leads to the expression of repressed genes and ultimately triggers differentiation, apoptosis, cell cycle arrest and cancer cell death through epigenetic or non-epigenetic means [76]. These results are especially promising in patients unsuitable for intense chemotherapy. Unfortunately, additional research is required before HDACis may be used as monotherapeutic agents against AML. In clinical studies, however, combination tactics with a range of anticancer medications are demonstrating strong antileukemic activity by augmenting the efficacy of traditional AML pharmacons [77]. Various mechanisms of action of HDACis in hematological cancers are depicted in Figure 3, and the HDACis landscape as it pertains to AML is summarized below.

In a study utilizing AML cells, researchers observed the effects of the new HDACi panobinostat (LBH589) in conjunction with doxorubicin [78] All AML cell lines studied, and patient-derived AML primary cells, showed a strong response to panobinostat (IC_50_ 20 nM). Panobinostat also enhanced the efficacy of doxorubicin and other first-line treatments for AML. Gene expression profiling analysis revealed that the application of panobinostat plus doxorubicin had a unique effect on the expression of 588 genes. Specifically, combination treatment upregulated Bax, and, most notably, Bad, which led to caspase-dependent apoptosis in AML cells via mitochondrial outer membrane permeabilization and cytochrome c translocation. The drug combination strongly activated a DNA damage response, suggesting it may generate DSBs and cause cell death. Continued research is important to optimize the pharmacological potential of panobinostat, alone or in conjunction with doxorubicin against AML [79]. Other reports implicated the interference with HSP90’s chaperone function in panobinostat’s antileukemia activity. C-X-C-chemokine receptor type 4 (CXCR4) is a receptor for stromal cell-derived factor 1, and a marker of poor prognosis when found in excess in leukemic cells. CXCR4 is also a client protein of HSP90. Application of panobinostat to AML cells was determined to induce HSP90 acetylation and diminish CXCR4 levels, possibly by inhibiting CXCR4-HSP90 interaction. Panobinostat was also found to be therapeutic in the treatment of mice harboring t(8;21) translocations via proteasomal degradation of AML1-ETO9a, the spliced variant of the t(8;21)-generated fusion protein known to induce leukemia [80].

### 8.2. Lymphomas

Lymphoma describes several forms of blood cancer with similar symptoms to leukemia, but with distinct etiology. Leukemia develops in the bone marrow and other blood-forming tissues, while lymphoma begins in the lymphatic system. The lymphatic system, which includes the bone marrow, spleen and lymph nodes, is a part of the immune system that assists in the defense against infection [81]. 

The two most common subtypes of lymphoma are Hodgkin’s (HL) and non-Hodgkin’s (NHL). The first is considered to be the more aggressive form and is defined histologically by large B-cells with “owl’s eyes” called Reed-Sternberg cells. NHL is up to nine times more common and has over 60 variants. Two groups of NHL are considered: B-cell lymphomas and natural killer/T-cell lymphomas. Also important for therapeutic management is to categorize NHL because some forms are more aggressive [82]. 

HDACis may play a therapeutic role in manipulating epigenetic and non-epigenetic events involved in lymphomagenesis, including the regulation of cytokines. Cutaneous T-cell lymphoma (CTCL), a subtype of NHL, is characterized by a high affinity IL-2 receptor in up to 30% of patients which may be compromised by HDACis. Additionally, HDAC1 and HDAC6 overexpression has been observed in CTCL, which results in increased production of IL-15—a critical mediator of inflammation in CTCL [83,84].

Peripheral T-cell lymphoma, another NHL subtype, was shown to be responsive to Tucidinostat, an HDACis which causes cell death in malignant cells by downregulating Bcl-2 and upregulating caspase-3 and Bax. HDACis were identified to influence the expression of key genes in CTCL cells. The Ras Homolog Family Member B (RhoB) is a tumor suppressor gene associated with different cancers. The combination of HDACi with conventional chemotherapy such as romidepsin, plus azacytidine, a demethylating agent, induces RhoB and apoptotic death, providing an effective treatment option for peripheral T-cell lymphoma [85].

T-cell malignancies are dysfunctional due to another self-consuming mechanism akin to apoptosis. In addition to deacetylating lysine residues on histones, HDACs regulate a range of cytosolic proteins with diverse biological roles, including autophagy. Vorinostat, a pan-HDACi, upregulated the autophagic factor LC3, blocking the mammalian target of rapamycin (mTOR), resulting in the initiation of the canonical route of autophagosome formation by ULK1 [86]. HDACis and associated autophagy are mandatory for cellular survival, therefore the use of HDACis seems to be a logical therapy for T-cell lymphomas. The primary action of HDACis may be to interfere with histone and chromatin remodeling, although acetylation of histone and other proteins can also induce DNA modifications, alter the expression of oncogenes or disregulate apoptosis and autophagy responses. Because these processes are potential targets in T-cell malignancies, they underline the importance of HDACis in cancer development and prognosis [87].

Cambinol is an inhibitor of the NAD-dependent deacetylase activity of human SIRT1 and SIRT2 and exhibited activity against Burkitt lymphoma. During genotoxic stress, cambinol inhibits SIRT1 activity, resulting in hyperacetylation of critical stress response proteins and encouraging cell cycle arrest. Hyperacetylation of B-cell lymphoma 6 (BCL6) and p53 accompanied apoptosis in BCL6-positive Burkitt lymphoma cells treated with cambinol alone. Cambinol’s anticancer action in Burkitt lymphoma might be achieved by a combination of BCL6 inactivation and checkpoint activation since acetylation inactivates BCL6 and promotes p53 and additional checkpoint signaling pathways. Moreover, cambinol decreased the development of model Burkitt lymphoma xenografts and was well tolerated by mice [88]. Other small molecule inhibitors sirtinol and M15, structural analogues of substituted β-naphthols, were discovered using high-throughput phenotypic screening in yeast [89]. Sirtinol was found to silence yeast Sir2 in vitro and in vivo, as well as human SIRT2 in vitro, but not HDAC1. Splitomicin, a compound structurally related to sirtinol, was discovered by a chemical screen of yeast cells and was found to inhibit the Sir2 deacetylase activity [90]. The deacetylation of key regulatory proteins BCL6 and p53 by NAD-dependent HDACs like SIRT1 has been linked to cell adaptations to cancer and stress.

### 8.3. Multiple Myeloma

Multiple myeloma (MM) is a hematological malignancy characterized by monoclonal plasma cells, most frequently producing IgG or IgA. Of the 1,278,362 new cases of leukemia, lymphoma, and MM in 2020 globally, 176,404 (14%) were MM. Greater rates of incidence and mortality were associated with higher levels of human development, income, frequency of inactivity, excess body weight, obesity, and diabetes. Global trends point towards an increase in MM incidence on a global level. Those at highest risk include men and individuals 50 years and older. Fortunately, mortality attributable to MM is decreasing, especially in women. Still, treatment availability and options should be improved to address the increasing incidence of multiple myeloma [91]. Autologous stem cell transplantation and innovative therapies, such as bortezomib (a proteasome inhibitor) and immunomodulatory medicines (lenalidomide, pomalidomide, thalidomide), have augmented the survival of MM patients. Numerous instances of relapsed myeloma have been documented, necessitating the development of novel medicines to treat these patients [92].

Multiple preclinical studies have shown the effectiveness of HDACis in the treatment of MM. The HDACis vorinostat promotes p21WAF1 in myeloma cells by altering the methylation and acetylation of core histones, and by reducing DNase I enzyme accessibility in promoter regions. By suppressing interleukin-6 release from MM-bound bone marrow stromal cells, vorinostat was also able to overturn cell adhesion-mediated drug resistance [93]. Additionally, MM cells pretreated with bortezomib were more susceptible to the apoptosis and mitochondrial dysfunction caused by vorinostat. Indeed, this combination was effective against MM cells resistant to both dexamethasone and doxorubicin [93]. 

The proteosome regulates the cell proteome clearing cytotoxic ubiquitinated proteins not correctly folded. MM cells generate an abundance of proteins, including immunoglobulins, and some misfolded/unfolded proteins, which impair cell functioning [94]. Relying on the proteasomal removal of harmful proteins, malignant cells are more vulnerable to mechanisms of proteasomal inhibition compared to healthy cells. When amounts of ubiquitinated misfolded/unfolded proteins exceed the proteasome’s destruction capabilities, these proteins collect aggresomes form. Aggresomes are microtubule-based pericentriolar structures, often generated in response to the excess of misfolded/unfolded proteins. The transport of aggresomes and capture of cytotoxic proteins is mediated by HDAC6. Therefore, inhibition of HDAC6 can hinder cells from properly forming aggresomes and prevent them from eliminating protein aggregates [95]. Bortezomib is a known inhibitor of the proteasomal chymotrypsin activity. Normally, bortezomib treatment stimulates aggresome development in MM cells. When combined with panobinostat, another pan-HDAC inhibitor, the treatment promoted apoptosis via a synergistic mechanism by preventing protein degradation. This may partially explain the synergistic effect of bortezomib and panobinostat in multiple myeloma patients [96]. 

The protein phosphatase 3 or calcineurin is another target of panobinostat. It is a calcium-dependent serine/threonine protein phosphatase and panobinostat induces degradation of its catalytic subunit α (PPP3CA). The phosphatase dephosphorylates NFATc1 and the subsequent translocation of cytoplasmatic NFATc1 to the nucleus is critical for the activation of T-cells [97]. Cyclosporine A and FK506, are commonly used immunosuppressives that function by blocking PPP3CA and calcineurin B. Calcineurin activation has been demonstrated to contribute to the maintenance of T-cell acute lymphoblastic leukemia in a mouse model. B-cell functionality is also linked to calcineurin activity. One study determined that CD138+ cells obtained from patients with advanced MM contained high levels of PPP3CA. This finding suggests a relationship between MM pathogenesis and PPP3CA (Figure 4). Interestingly, PPP3CA acted as a client protein of HSP90 in MM cells [98].

Ricolinostat (or ACY-1215) is an orally active hydroxamate and was the first selective inhibitor of HDAC6 in clinical trials [98]. Therapy with ricolinostat prompted the destruction of PPP3CA by dissociation with HSP90, indicating panobinostat may reduce levels of PPP3CA by abolishing HSP90’s chaperone function. Moreover, PPP3CA was vital for MM cell growth. While no changes in PPP3CA levels were observed following FK506 treatment, application of this compound with panobinostat did result in the reduction of PPP3CA levels, as well as growth inhibition in preclinical trials. Cotreatment is believed to result in PPP3CA loss via interference with calcineurin B function, which is usually responsible for its protection [99].

A combination of panobinostat with dexamethasone and bortezomib seemed efficacious in patients harboring bortezomib-resistant disease (Figure 5). A review of many patients with MM showed that those with bortezomib-resistant disease had very high levels of PPP3CA. In practice, bortezomib caused inhibition of HDAC6 and reduced expression of PPP3CA. Furthermore, combination therapy with panobinostat plus bortezomib synergistically reduced the viability of MM cells. So, both panobinostat and bortezomib target PPP3CA and the antimyeloma effects of their combination may result from the synergistic PPP3CA reduction, besides aggresome inhibition by bortezomib [98]. Ricolinostat also demonstrated reasonable effectiveness against relapsed MM combined with dexamethasone and lenalidomide proteasome inhibitor [98].

## 9. Development of HDAC Inhibitors for Hematological Cancer Treatment

All approved HDACis are presently being tested in clinical trials for the treatment of additional cancer settings, alone or combined in adjuvant regimens. Numerous advancements have been achieved in the treatment of solid neoplasms, such as brain and pulmonary cancers, as well as hematological malignancies like AML and B-cell lymphoma. Vorinostat is the subject of nearly 60 active clinical trials, including three phase 3 trials evaluating drug combinations in glioma (NCT01236560), multiple myeloma (NCT01554852), and acute lymphoblastic leukemia and lymphoma (NCT01312818). Tucidinostat could be found in more than 50 studies at ClinicalTrials.gov with various hematological malignancies. Panobinostat is the subject of over 20 trials, including one phase 3 trial aiming at the treatment of AML and myelodysplastic syndromes [99].

Researchers have discovered 22 HDACis that entered clinical trials against different cancers (Table 4). The hydroxamate family is the more represented (11 compounds), but diverse non-hydroxamate drugs are also investigated. Most of these medicines are now being evaluated for hematological cancers, although their effectiveness and safety profiles differ. The creation of isozyme-selective inhibitors is one of the most intriguing areas of research. In the phase 1b study with Epstein–Barr virus (EBV)-associated lymphomas, the HDAC9-selective medication nanatinostat showed promising results, with an overall response rate of 53% and 29% of patients achieving a complete response. In this trial, HDACis appear to work by sensitizing EBV-positive lymphoma cells to nucleoside antiviral drugs like ganciclovir [100].

## 10. Natural Products as HDAC Inhibitors in Hematological Malignancies

### 10.1. Pure Compounds

#### 10.1.1. Berberine 

Berberine (Figure 6), an isoquinoline alkaloid isolated from the plant *Berberis aristata*, has been studied in depth for its anticancer effects in various malignancies [101,102,103]. Berberine was also demonstrated to hinder cancer spread by inhibiting transferase activity. It has been hypothesized that berberine acts by inhibiting HDAC proteins [102]. Initially, molecular docking investigations were conducted using Glide (Schrodinger) and Sitemap (Schrodinger) was used to forecast the live website. The ligands of berberine exhibited a strong affinity for the HDAC family (PDB IDs 2VQM, 3C10, 3C5K, 3SFF, 3MAX, 4A69, and 4BKX). Recent research indicates the probable molecular target of berberine utilizing the Connectivity Map (CMap) database and a gene expression signature-based method. It is hypothesized that berberine might inhibit protein synthesis, HDACs, and Akt/mTOR pathways, which was highly correlated with in silico predictions (Table 5) [103]. Despite preliminary findings, there is currently no reliable evidence of berberine’s underlying molecular mechanism for tumor targeting [103].

#### 10.1.2. Chrysin

The flavonoid chrysin (Figure 6) occurs in propolis, *Oroxylum indicum* and *Pelargonium crispum*, and has shown potential antitumor action against melanoma and leukemia [104,105]. Research suggests chrysin acts as an epigenetic modulator by inhibiting specific HDACs. Chrysin, at a concentration of 40 μM, inhibited HDAC8 activity, and also decreased the protein levels of HDAC8 and HDAC2 in vitro. Chrysin was tested on HDAC1-11 and found to selectively inhibit these two isoenzymes as well. Based on the EC_50_ values, chrysin is a more potent inhibitor of HDAC8 than of HDAC2 (40.2 μM versus 129.0 μM). Chrysin has also demonstrated anticancer action against blood malignancies. Chrysin’s antiproliferative action on leukemic cells followed a concentration-dependent pattern. Chrysin caused apoptosis through the mitochondrial route in peripheral blood cells isolated from patients with chronic lymphocytic leukemia. Importantly, fibroblasts and epithelial cells were resistant to the cytotoxicity generated by chrysin [106,107].

#### 10.1.3. Cowaxanthone and Cowain

*Garcinia fusca* Pierre, also known as “Madan-Paa” or “Mak-Mong”, is a member of the Clusiaceae family and native to northeastern Thailand and other southeast Asian nations. Traditional applications of *G. fusca* include uses for blood circulation enhancement, as an expectorant, or as therapy for coughs, indigestion, constipation, and pyrexia [108]. Two active chemicals isolated from *G. fusca* have been evaluated for their therapeutic potential in the context of hematological malignancies. Cowaxanthone (Figure 6), one of the xanthone families derived from *G. fusca*, displayed anticancer properties by inducing apoptosis in leukemic T-cells via downregulation of HDACs activity [109]. However, cowaxanthone was less hazardous to normal Vero cells. Cowanin (Figure 6) is a xanthone isolated from the stem bark of *G. fusca*. Cowanin has been shown to inhibit the growth of a variety of cancer cells, including those derived from lung, breast, oral, and liver cancers, in addition to leukemic HPB-ALL cells [109]. Moreover, cowanin has been purported to suppress the Notch signaling system, which is characteristically overactive in hematologic cancers. Cowanin’s activity as an HDACis is supported by its stimulation of cancer cell death in Jurkat cells (human T-cell lymphoma) via apoptosis and autophagy [110]. Thus, cowanin shows promise as an anticancer drug, and its mechanism merits further study.

**Figure 6 cancers-15-02808-f006:**
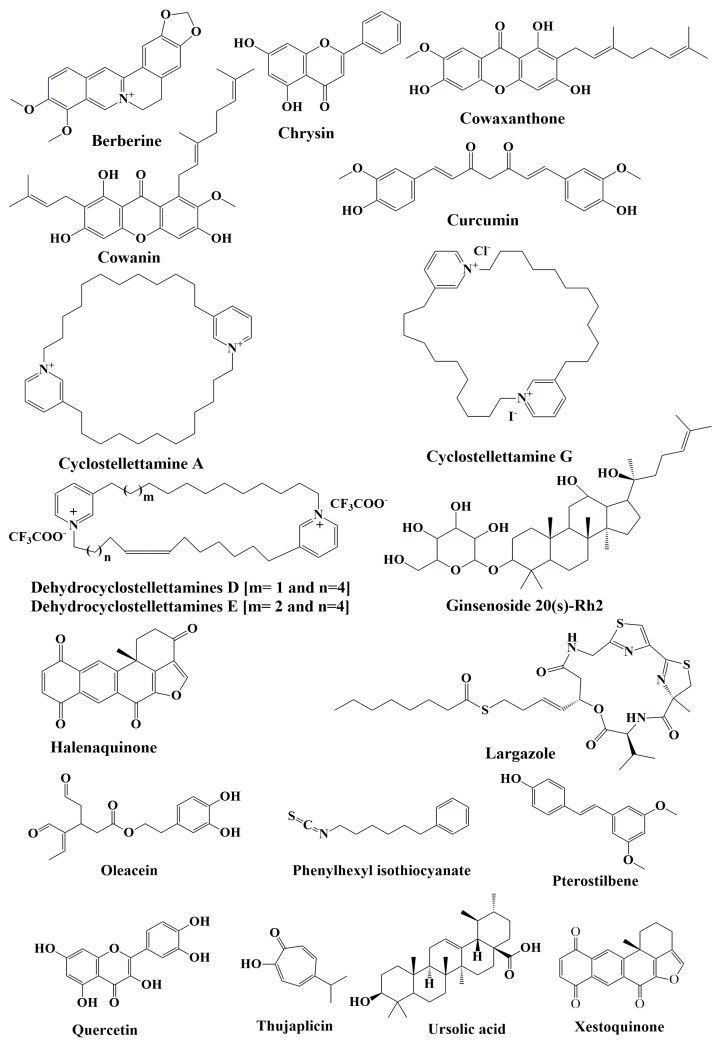
Chemical structure of natural compounds acting as HDAC inhibitors for hematological cancers.

#### 10.1.4. Curcumin 

Curcumin (diferuloylmethane, Figure 6), a lipid-soluble polyphenol derived from the *Curcuma longa* rhizome, is possibly one of the most studied natural anticancer agents [111,112]. The chemo preventive activity of curcumin, in part, is due to its role in epigenetics. In leukemic cells, curcumin inhibits HDAC4 and HDAC6 [113]. Moreover, it increased histone acetylation at the promoter of the gene of the proapoptotic protein Bax, in association with activity inhibition and decreased expression of different HDACs. Despite these findings and its proven applications in other cancer models, curcumin has been understudied in the context of hematological malignancies.

#### 10.1.5. Cyclostellettamines 

Cyclostellettamines are pyridine alkaloids that have been isolated from marine sponges (e.g., genus *Stelletta* and *Xestospongia*). Figure 6 presents the structures of cyclostellettamines A and G, as well as dehydrocyclostellettamines D and E. The ability of these compounds to inhibit the histone deacetylase activity was tested with K562 cell preparations and IC_50_ values were measured in the interval of 17–80 μM [114]. When tested against leukemia cells, these compounds exerted moderate cytotoxic effects [115]. 

#### 10.1.6. Ginsenoside 20(s)-Rh2

Ginseng, an herbal medicinal from the genus *Panax*, has numerous pharmacological activities. Ginsenosides are triterpene saponins, regarded as the primary active components present in the roots and rhizomes of various *Panax* plants [116]. Specifically, ginsenosides have displayed anticancer, anti-inflammatory, antioxidative, and vasorelaxant properties [117]. Among the diversity of ginsenosides, 20(s)-Rh2 (Figure 6) was demonstrated to induce cell cycle arrest and apoptosis of a variety of human cancer cells. Recent research demonstrates that 20(s)-Rh2 can improve the response of human leukemia cells to conventional therapies and reduce cell proliferation. Researchers postulate these results to be due to upregulation of mitochondria-mediated apoptosis, as well as immunomodulatory effects [117]. 

Chen et al. [118] examined the 20(s)-Rh2 actions in K562 and KG-1a cultured cells using immunoblotting to determine levels of protein expression. In association with cell cycle arrest (G0/G1 phase) and apoptosis, the ginsenoside reduced proliferation-related proteins (Bcl-2, ERK, cyclin D1, and p-ERK), and augmented pro-apoptotic factors (activated caspase-3, Bax, JNK and p-JNK, p38 and p-p38). Moreover, 20(s)-Rh2 inhibited HDAC1, 2 and 6, while increasing the acetylation of histone H3 and HAT activity. In summary, 20(s)-Rh2 slowed the proliferation of leukemia cell models while modulating HDACs activity, histone acetylation, and critical cell fate proteins. These effects have been validated in a follow-up in vivo study utilizing naked mice carrying K562. The authors found that rodents treated with 20(s)-Rh2 had smaller xenografted tumors.

#### 10.1.7. Halenaquinones

Halenaquinone (Figure 6), a polycyclic quinone-type metabolite produced by *Xestospongia vansoesti* and *Paracheilinus alfiani*, has been shown to inhibit pan-HDACs and DNA replication via suppression of topoisomerase II production. Anticancer effects of halenaquinone have been observed against Molt-4 leukemia cells, in which experimental results showed reduced cell viability and induction of apoptosis following ROS-induced mitochondrial dysfunction [119]. Researchers attribute these findings to upregulation of Bax, cytochrome c and hexokinase I release, and activation of caspases, with concomitant downregulation of Bid, Bcl-2, cytosolic hexokinase II, p-Akt, tensin homolog, phosphatase, phosphoinositide-dependent kinase-1, and glycogen synthase kinase-3β. Notably, halenaquinone showed a strong in vivo antileukemic impact in mouse xenograft experiments, decreasing the tumor mass and volume without altering the overall body mass of the mice [120].

#### 10.1.8. Largazole

The cyclic depsipeptide largazole (Figure 6) was extracted from Key Largo (Florida) marine cyanobacterium of the genus Symploca in 2008 [121]. The synthesis of largazole and its C7-substituted analogues (4c-d), and the evaluation of these compounds in NB4 leukemia cells, was reported by Souto et al. [121]. The largazole analogues inhibited HDAC1 and 4 recombinant enzymes, which are models of class I and class II HDACs. These compounds were found to possess nanomolar potency against HDAC1. In addition, they exhibited potent apoptosis-inducing action in the NB4 cells, whereas were incapable of inducing their differentiation into mature granulocytes. After treating the leukemia cells with the compounds, increased levels of acetylated tubulin and p21^WAF1/CIP1^ were observed, besides higher global acetylation of H3.

#### 10.1.9. Oleacein 

The secoiridoid oleacein (Figure 6) is a polyphenol richly present in *Olea europaea* (common olive). This compound was demonstrated to modulate cancer cells through alterations in proliferation, apoptosis, and differentiation in several different models, including colon, breast, prostate, and skin cancers. Reports indicate that oleacein regulates epigenetic mechanisms in diverse myeloma cell models, namely NCI-H929, RPMI-8226, U266, MM1s, and JJN3. Showing micromolar IC_50_ values, oleacein diminished the viability of myeloma cells in 48-h incubations [122]. 

Safety assays of oleacein with healthy human peripheral blood mononuclear cells (PBMC) revealed no toxicity. Unlike the positive controls, TSA and SAHA, oleacein had no effect on the HDAC activity of JJN3 nuclear extracts. However, it downregulated the expression of several HDACs and increased histone and tubulin acetylation, without changes in the global DNA methylome [123].

#### 10.1.10. Phenylhexyl Isothiocyanate 

Isothiocyanates found in plants and synthesized phenylhexyl isothiocyanate (PHI, Figure 6) inhibit carcinogen-induced tumor formation in rodents [124]. Ma et al. [124] found that the chemopreventive isothiocyanates can modulate the growth, upregulating p21^WAF1^ inhibitor and suppressing the activity of CDK, important cell cycle regulators in cancer cells. In this study, it was observed that PHI exposure of HL-60 leukemia cells caused cell cycle arrest at G1 and induction of apoptosis. 

Researchers investigated the mechanism by which PHI suppressed the proliferation of cancer cells, ultimately attributing this phenomenon to its role as an HDACis. In addition, PHI decreased the HDAC expression and raised the amount of the acetyltransferase p300, causing an increase in acetylated histone accumulation. PHI also led to particular alterations to histone methylation, suggestive of favorable transcriptional activity. Chromatin immunoprecipitation assay revealed increased p21 DNA in the hyperacetylated histones from the PHI-exposed cells. This finding indicates chromatin unfolding leads to heightened access of the transcriptional elements to the location of the p21 promoter, resulting in the activation of cell cycle inhibitors. Apoptosis of HL-60 cells triggered by PHI was modulated by caspase-9 and Bcl-2, but normal PBMC or those obtained from the bone marrow were not susceptible to the compound. Therefore, PHI may act selectively on cancer cells [124].

#### 10.1.11. Pterostilbene

Another polyphenol, pterostilbene is a stilbenoid (Figure 6) found primarily in blueberries, grapes, and wood. Pterostilbene is a natural dimethylated analog of resveratrol possessing greater liposolubility and bioavailability, making it possibly a more effective anticarcinogenic molecule [125]. Pterostilbene induces apoptosis in diverse malignancies—bladder, breast, colon, lung, pancreatic, and stomach cancers, as well as leukemia cells [126,127]. Naturally-derived resveratrol analogues, pterostilbene and 3′-hydroxypterostilbene have been shown to be efficient apoptosis-inducing drugs in multiple-drug resistance (MDR) and BCR-ABL-expressing leukemia cells. Different in vitro analysis in bortezomib resistance cell line H929R underscores their significance in the treatment of resistant hematologic malignancies through cell cycle arrest, autophagy, and apoptosis [127].

#### 10.1.12. Quercetin

Quercetin (Figure 6) is a flavonol present in many plants, including various fruits, vegetables, and grains as well as leaves and seeds with red apples, onions, parsley, and sage are popular foods containing substantial levels of quercetin [128]. Alvarez et al. [129] explored the molecular processes behind quercetin’s anticancer actions by analyzing its impact on DNA methylation and histone modification. This research was conducted in vivo utilizing mice with human AML tumor xenografts and in vitro on HL-60 and U937 cells. Incubations with quercetin significantly down-regulated the expression of DNMT1 and DNMT3a, an effect partially reliant on STAT-3. Class I HDACs were likewise downregulated by the therapy. Additionally, co-treatment with quercetin plus the proteasome inhibitor MG132 reduced the loss of class I HDACs in comparison to quercetin alone, demonstrating that quercetin promoted HDACs degradation by proteasomes. The methylation of the proapoptotic BCL2L11 and DAPK1 genes was decreased by quercetin in a concentration- and time-dependent manner. In addition, treatment of cells with the flavonoid, for 48 h, buildup acetylated H3 and H4 in the promoters of BNIP3L, BAX, BCL2L11, DAPK1, APAF-1, and BNIP3L. In accordance, treatment with quercetin substantially enhanced the transcription of each of these genes. The accumulation of H3ac and H4ac at the promoters of genes implicated in apoptosis, resulting in their increased expression, may contribute to accelerated apoptosis induced by quercetin, according to the findings.

#### 10.1.13. Thujaplicin 

Thujaplicins (Figure 6) are naturally occurring substances generated from tropolone, a toxin produced by the agricultural pathogen *Burkholderia plantarii*. Recently, Ononye et al. [130] revealed that some tropolones potently and specifically target HDACs and impede the proliferation of hematological cell lines. A naturally occurring non-benzenoid aromatic containing a metal-directing tropone unit comprises the tropolone nucleus. Tropolones show an antiproliferative effect on Jurkat cells in a way unique from vorinostat, a pan-HDAC inhibitor. While both agents cause apoptosis in this cell line, it seems that the tropolones function via a smaller selection of biological processes. Specifically, compared to vorinostat, tropolones had a lesser impact on histone hyperacetylation and do not seem to increase the expression of p21. Vorinostat appears to activate an extrinsic mechanism of apoptosis; the tropolones do not, although both enhance the activity of caspase-3 and caspase-7. Both tropolones and vorinostat seem to increase the expression of perforin, which may be associated with cell maturation and, perhaps, caspase-3 activation. Overall, the more restricted activation of apoptosis by tropolones compared to pan-HDAC inhibitors may enable the creation of more targeted anti-leukemia therapies. In this sense, the tropolone nucleus is well-suited for producing HDAC inhibitors with specific cellular effects, which may lead to the development of drugs with unique therapeutic properties.

#### 10.1.14. Ursolic Acid

There are several plants and foods that contain ursolic acid (Figure 6), a pentacyclic triterpenoid [131]. Some natural sources of ursolic acid include rosemary (*Rosmarinus officinalis*), ginseng (*Panax ginseng*), plum (*Prunus domestica*), pear, apple peel, and cranberry. Its potential for both cancer prevention and treatment has been investigated. Studies on ursolic acid have focused on its potential to inhibit HDAC proteins; HDAC1-3 and 6-8 were shown to be significantly downregulated by this bioactive chemical, beginning at a concentration of 2.5 μM (a non-toxic concentration) [132]. Moreover, in leukocytes, ursolic acid counteracted HDAC1 and HDAC3 induction.

#### 10.1.15. Xestoquinone 

The polycyclic quinone-type metabolite termed xestoquinone (Figure 6) is also a marine drug isolated from sponges of the *Xestospongia* and *Petrosia* genus, and has been shown to have an anticancer potential [133]. 

The cytotoxic effects of xestoquinone on various hematological cancer cell lines have been investigated in preclinical trials. In one study, cell growth was inhibited by the compound with low-micromolar IC_50_ values. Additionally, topoisomerase I and II activity were reduced by 50% and 30%, respectively, at concentrations of 0.235 and 0.094 μM in the cell-free system. Researchers purport xestoquinone’s lethal action on Molt-4 cells to be caused by the activation of many distinct apoptotic pathways, such as the death-receptor pathway. Pretreatment of these cells with N-acetyl cysteine alleviated the loss of mitochondrial membrane potential, decreased apoptosis, and maintained topoisomerase I and II expressions. In another study, xestoquinone (1 μg/g) effectively inhibited leukemia cancer cell development in a nude mouse xenograft model, decreasing it by 31.2% relative to the solvent control [115]. These results warrant further investigation into xestoquinone as a potential antileukemic agent capable of targeting cancer cells via numerous pathways [115]. 

### 10.2. Plant Products and Extracts

#### 10.2.1. *Antrodia camphorate*

Due to its antioxidant properties, the plant *Antrodia camphorata* is widely used in herbal medicine in Taiwan. The capacity of *A. camphorata* to induce apoptotic cell death in a leukemia model was investigated by Yang et al. [134]. Using the HL-60 cell line, the authors observed a decrease in cell viability, accompanied by chromatin condensation and internucleosomal DNA fragmentation indicative of apoptosis, following treatment with 20–80 µg/mL A. camphorate extract. Researchers also note an increase in cytochrome c release, caspase-3 activation, and the proteolytic cleavage of PARP. An uptick in *A. camphorata*-induced apoptosis coincided with a decrease in Bcl-2, a powerful inhibitor of cell death, and upregulation of Bax [135]. Accordingly, *A. camphorata* may possess anticancer potential, and so further studies are warranted to elucidate its role against hematological malignancies. In support of these findings, follow-up in vivo analysis in which investigators observed delayed tumor volume in a xenograft mice model treated with 80 or 120 mg/kg *A. camphorata* extract. Tissue analysis confirmed these results to be mediated by increased cell cycle arrest [135]. 

#### 10.2.2. *Feijoa sellowiana*

*Feijoa sellowiana* Berg. (Myrtaceae) is a bushy evergreen shrub native to South America that has adapted well to the Mediterranean climate. *F. sellowiana* grows all over the Mediterranean and is widely consumed as a food source. *F. sellowiana* also contains high levels of bioflavonoids or vitamin P-active polyphenols, including catechin, leucoanthocyanins, flavonols, proanthocyanidins, and naphthoquinones [136]. It was reported that the acetonic extract of *F. sellowiana* selectively inhibited the growth of hematological cancer cells when applied to malignant cells and healthy myeloid progenitors. Flavone (2-phenyl-4H-chromen-4-one) was determined to be the active component in the *F. sellowiana* acetonic extract through fractionation, purification, and analysis. Flavone induces apoptosis in human myeloid leukemia cells, which is accompanied by caspase activation and increased expression of p16, p21, and TRAIL. The use of ex vivo myeloid leukemia blasts confirms the ability of both the full acetonic *F. sellowiana* extract and its derived flavone to induce apoptosis. In myeloid leukemia patient blasts and in other cell models, the apoptotic inducing action of *F. sellowiana* extract and flavone is accompanied by HDAC inhibition and associated rise in acetylation of histone and other proteins. These findings demonstrated for the first time that the apoptotic active principle of *F. sellowiana* is flavone, whose effects are at least in part attributable to its impact on acetylation status. These findings support the hypothesis that its epigenetic pro-apoptotic regulation in hematological cancers is dependent on HDAC inhibition [136].

#### 10.2.3. Olive Oil 

Mutations in somatic cells are crucial to cancer development and progression. Environmental and endogenous genotoxic chemicals induce genetic changes. Reactive oxygen species, which are produced in all aerobic organisms as metabolic byproducts and also as bactericidal agents in phagocytic cells, are likely to be the most important endogenously produced genotoxic chemicals [137]. Thus, oxidative stress has been linked to several degenerative disorders, including cancer. 

Due to their antioxidant properties, small phenolic chemicals in olive oil are thought to prevent cancer. The impact of olive oil bioactives on DNA damage by H_2_O_2_ incubation of human PBMC and leukemia HL-60 cells has been examined previously [138]. Hydroxytyrosol and a combination of polyphenols derived from olive oil and olive mill residues decreased DNA damage at concentrations of 1 μM. With 10 μM hydroxytyrosol, 93% viability of HL-60 cells and 88% of PBMC cells were protected. In both types of cells, the dialdehydic version of elenoic acid coupled with hydroxytyrosol exhibited a comparable protective effect. Other isolated substances, like tyrosol, oleuropein and verbascoside similarly attenuated the DNA damage. Overall, these data imply that olive oil and its bioactive constituents can be useful in preventing the starting phase of carcinogenesis in vivo, since the quantities necessary to prevent oxidative DNA damage are readily attained with typical olive oil consumption [138].

**Table 5 cancers-15-02808-t005:** Natural anticancer HDAC inhibitors in hematological cancers.

Active Constituents	Source	Mechanism(s) of Action	References
Berberine	*Berberis aristate*	Inhibited protein synthesis, HDACs, and Akt/mTOR pathways	[101,102,103]
Chrysin	*Oroxylum indicum* and *Pelargonium crispum*	Derepressed the epigenetically silenced genes and inhibited HDAC8 both directly and indirectly by reducing its protein concentration	[104]
Cowaxanthone and Cowain	*Garcinia fusca*	Induced apoptosis and autophagy in leukemic T-cells	[109,110]
Curcumin	*Curcuma longa*	Increased histone acetylation on gene promoters of the proapoptotic BAX gene due to inhibition of HDAC1, HDAC3, and HDAC8 activity and expression in leukemic cells	[113]
Cyclostellettamine G and dehydrocyclostellettamines D and E	*Haliclona* and *Xestospongia*	Induced apoptosis	[114,115]
Ginsenoside 20(s)-Rh2	*Panax ginseng*	Induced apoptosis	[118]
Halenaquinone	*Xestospongia vansoesti* and *Paracheilinus alfiani*	Induced apoptosis	[120]
Hydroxytyrosol [3,4-dyhydroxyphenyl-ethanol (3,4-DHPEA)]	Virgin olive oil and olive mill wastewater	Induced DNA damage	[138]
Largazole	*Symploca* sp.	Induced apoptosis	[121]
Oleacein	*Olea europaea*	Exhibited epigenetic modulation in multiple myeloma cells	[122]
Phenylhexyl isothiocyanate	Cruciferous vegetables	Promoted G1 arrest and apoptosis	[124]
Pterostilbene	Blueberries and grapes	Cell cycle arrest, autophagy	[127]
Quercetin	Apples, onions, parsley, and sage	Induced apoptosis	[129]
Thujaplicin	Troplone	Promoted cell cycle arrest and induced apoptosis	[130]
Ursolic acid	*Panax ginseng*, *Rosmarinus officinalis,* and *Prunus domestica*	Promoted cell cycle arrest and induced apoptosis	[132]
Xestoquinone	*Petrosia* sp.	Induced apoptosis	[115]
Acetonic extract and flavone	*Feijoa sellowiana*	Induced apoptosis	[136]
Antrodia camphorate	*Antrodia camphorate*	Induced apoptosis	[135]

Abbreviations: HDAC, histone deacetylase; mTOR, mammalian target of rapamycin.

## 11. Conclusions

Epigenetic changes carried out by HDAC proteins are implicated in carcinogenesis and metastasis of hematological malignancies. Some HDACis are approved for the treatment of hematological malignancies, and many others are currently being investigated for their anticancer effects as monotherapies or adjuvant agents. HDACis have demonstrated effectiveness in AML, CTCLs, and MM by various mechanisms. These anticancer effects are broad in nature and are not limited to changes in acetylation status. Multiple preclinical studies demonstrate the role of HDACis as regulators of the cell cycle and agents of apoptosis. Current data most strongly supports the use of HDACis as adjuvants to conventional treatments. One study, for example, demonstrated the efficacy of panobinostat for therapy of refractory MM when used in combination with bortezomib and dexamethasone—even after a failed response to bortezomib monotherapy. The success of synthetic HDACis has fostered an interest in naturally occurring alternatives. Phytochemical HDACis, such as chrysin, oleacein, quercetin, as well as marine-derived HDACis halenaquinone and xestoquinone are being investigated as potential drugs against hematological cancers. Importantly, phytochemicals are generally less toxic compared to conventional chemotherapy, but their efficacy may be limited by bioavailability. Despite preliminary encouraging results, further studies are required to investigate the exact role HDACis play against hematological malignancies. Additionally, expanded in vivo research is required to elucidate the safety profiles and pharmacokinetics of these bioactive compounds before more robust clinical trials can ensue.

## Figures and Tables

**Figure 1 cancers-15-02808-f001:**
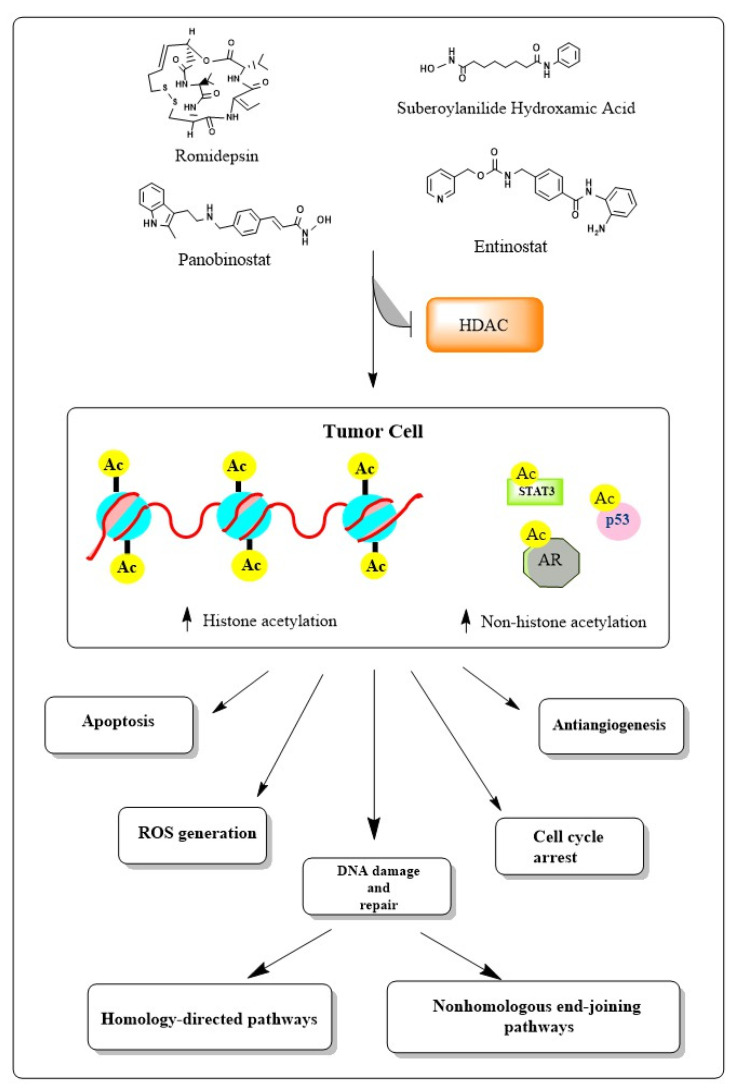
Schematic representation of methods of HDAC inhibition, inducing DNA damage.

**Figure 2 cancers-15-02808-f002:**
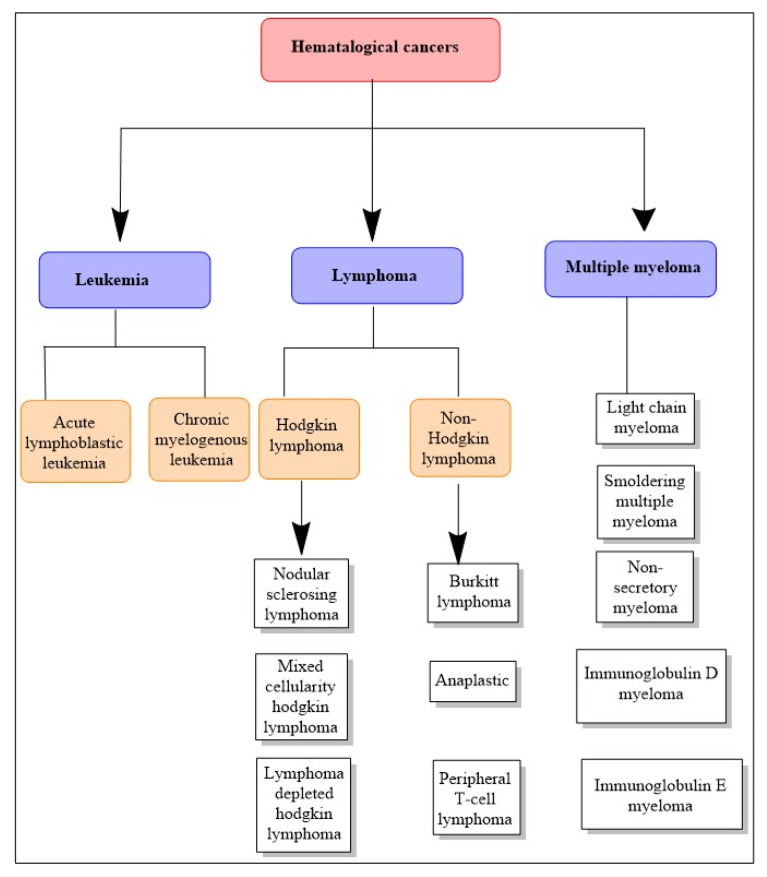
Schematic representation of the diversity of hematological cancers, with interrelated types and their subtypes displayed.

**Figure 3 cancers-15-02808-f003:**
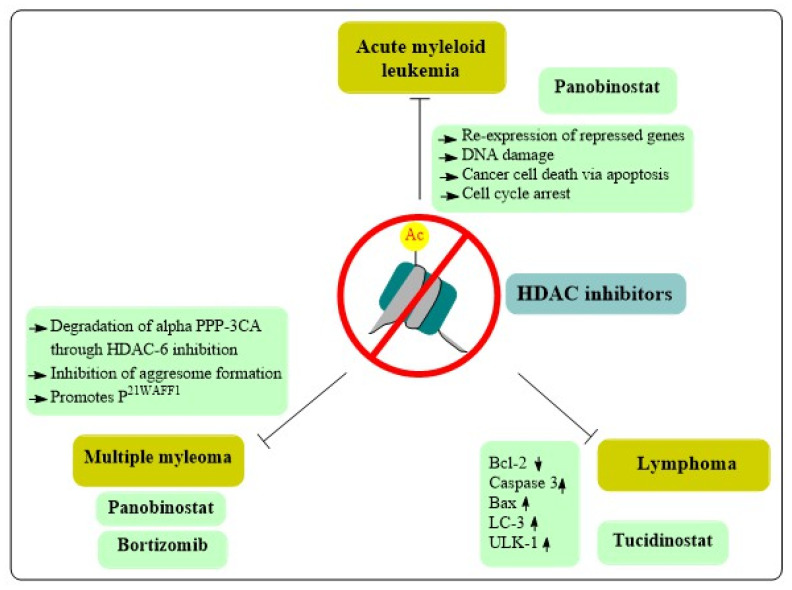
A summary of various mechanisms of action utilized by HDAC inhibitors in hematological cancers: disrupting tumor cell proliferation and survival through multiple mechanisms, including inducing DNA damage and affecting cell cycle progression.

**Figure 4 cancers-15-02808-f004:**
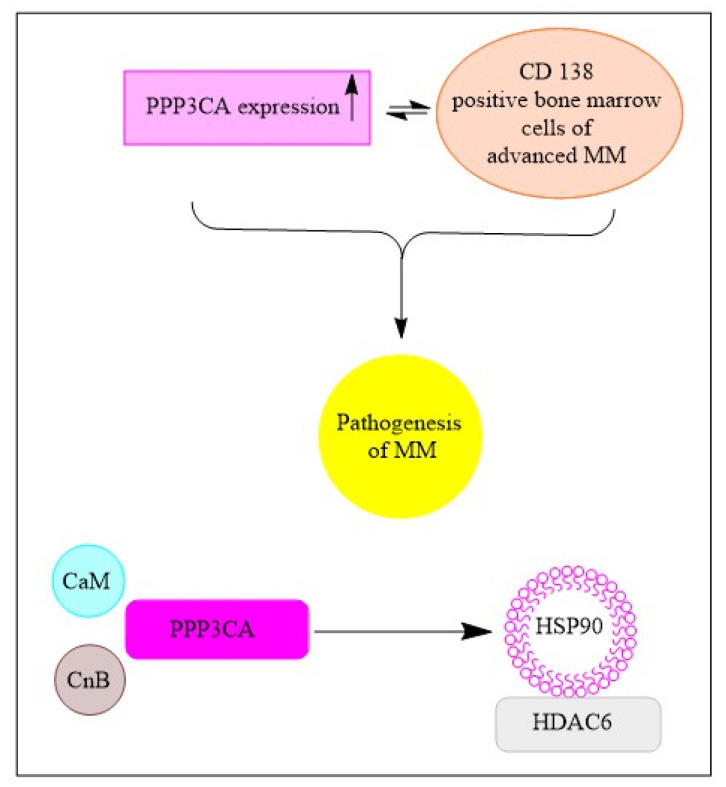
Schematic diagram indicating relationship of PPP3CA and pathogenesis of multiple myeloma before treatment.

**Figure 5 cancers-15-02808-f005:**
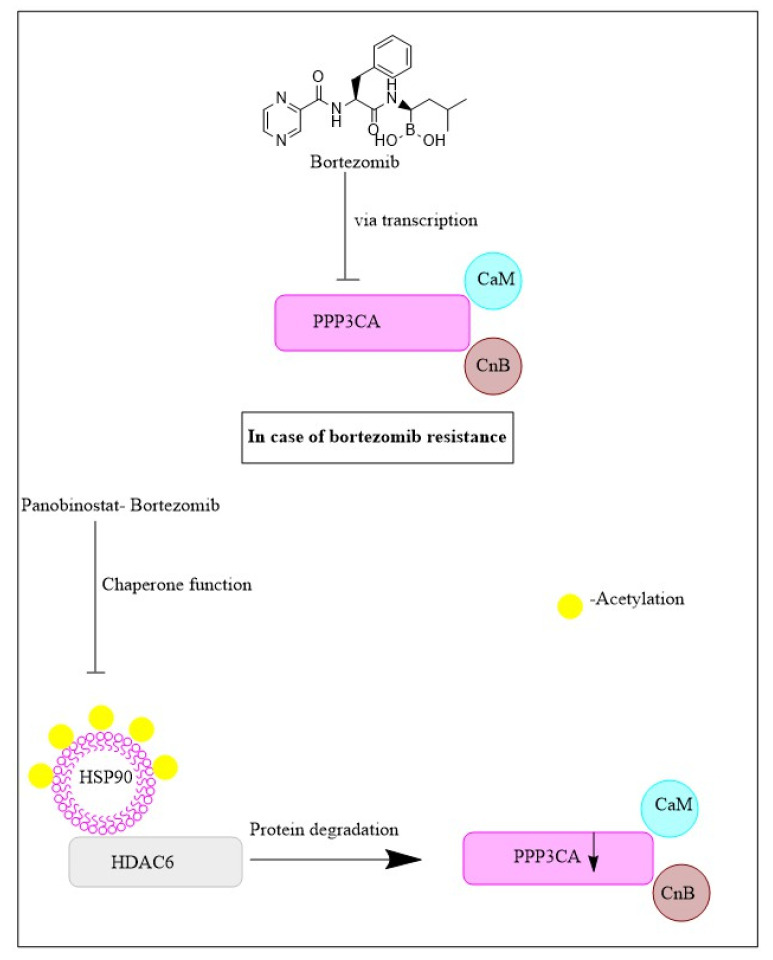
Schematic diagram of mechanism of action of panobinostat and bortezomib against multiple myeloma.

**Table 1 cancers-15-02808-t001:** Classification of HDACs with their cellular locations and physiological functions.

Class	HDAC	Localization	% Similarity to Human (Nucleic Acid/Amino Acid)	Associated Cancer(s)	Physiological Functions
I	HDAC 1	Nucleus	90.8/99.4	Prostate, gastric, colorectal, pancreas, and esophageal	Cell survival and proliferation
	HDAC 2	Nucleus	91.1/98.6	Colorectal, gastric, cervical dysplasia, and invasive carcinoma	Cell proliferation; insulin resistance
	HDAC 3	Nucleus	92.5/99.6	Lung, prostate, and colon	Cell survival and proliferation
	HDAC 8	Nucleus	90.9/96.3	Poor outcomes in pediatric neuroblastoma	Cell proliferation
IIa	HDAC 4	Nucleus/cytoplasm	86.3/94.2	Breast	Regulation of skeletogenesis and gluconeogenesis
	HDAC 5	Nucleus/cytoplasm	91.1/95.6	Colon, AML, and poor outcomes in lung cancer	Cardiovascular growth and function; gluconeogenesis; cardiac myocyte and endothelial cell function
	HDAC 7	Nucleus/cytoplasm	86.8/90.3	Colon	Thymocyte differentiations; endothelial function; gluconeogenesis; homologous recombination
	HDAC 9	Nucleus/cytoplasm	90.394.8	Medulloblastoma and astrocytoma	Thymocyte differentiation; cardiovascular growth and function
IIb	HDAC 6	Cytoplasm	81.1/78.7	Ovarian and AML	Cell motility; control of cytoskeletal dynamics
	HDAC 10	Cytoplasm	78.1/76.4	Hepatocellular carcinoma	Homologous recombination; autophagy-mediated cell survival
IV	HDAC 11	Nucleus/cytoplasm	87.3/91.9	Breast	Immunomodulators DNA replications
III	SIRT1	Nucleus	Not available	AML, colon, prostate, skin, and BCLL	Aging; redox regulation; cell survival; autoimmune system regulation
	SIRT2	Cytoplasm	Not available	Glioma	Cell survival-cell migration and invasion
	SIRT3	Nucleus/Mitochondria	Not available	Breast, prostate, head and neck, and glioblastoma	Urea cycle; redox balance; ATP regulation; metabolism; apoptosis; cell signaling
	SIRT4	Mitochondria	Not available	Breast	Energy metabolism; ATP regulation; metabolism; apoptosis; cell signaling
	SIRT5	Mitochondria	Not available	Pancreas and breast	Urea cycle; energy metabolism; ATP regulation; metabolism; apoptosis; cell signaling
	SIRT6	Nucleus	Not available	Colon and breast	Metabolism
	SIRT7	Nucleus	Not available	Breast	Apoptosis

Abbreviations: AML, acute myeloid leukemia; ATP, adenosine triphosphate; BCLL, B-cell chronic lymphocytic leukemia; HDAC, histone deacetylase; SIRT, sirtuin.

**Table 2 cancers-15-02808-t002:** Different classes of HDAC inhibitors with their cancer specificity.

Class	HDACis	Target	Structure	Associated Cancer(s)	References
Hydroxamates	SAHA (vorinostat)	HDAC1-11	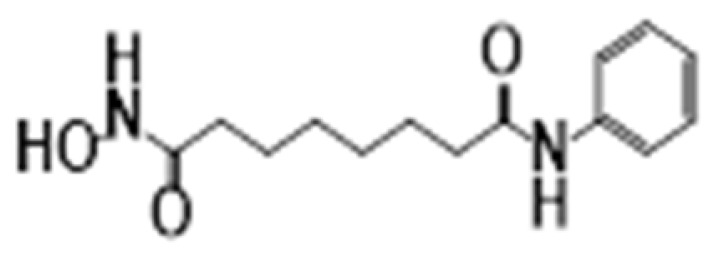	Glioblastoma multiforme and brain metastasis	[28]
	Panobinostat	HDAC1-11	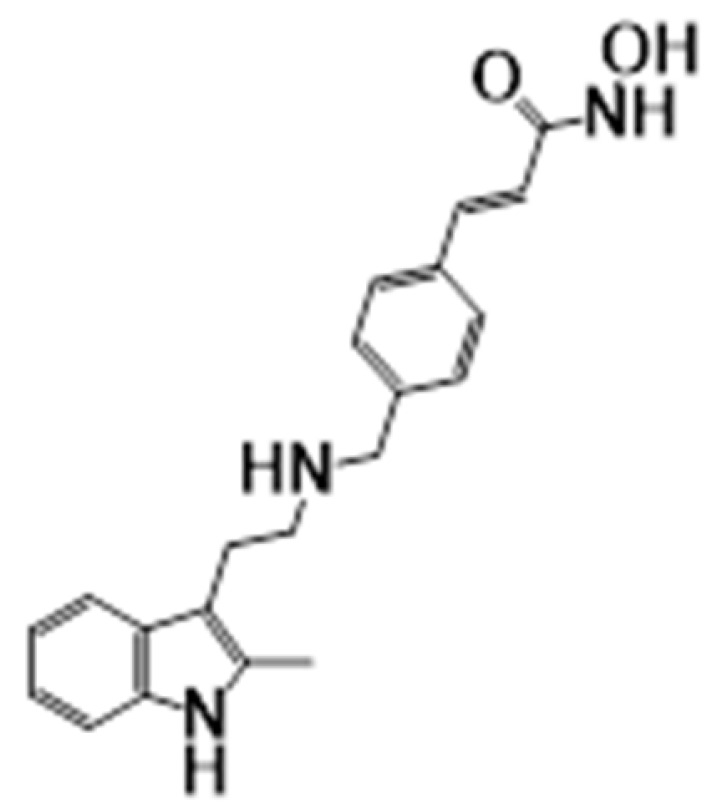	Small cell lung cancer, myelofibrosis, and cutaneous T-cell lymphoma	[29,30]
	Belinostat	HDAC1-11	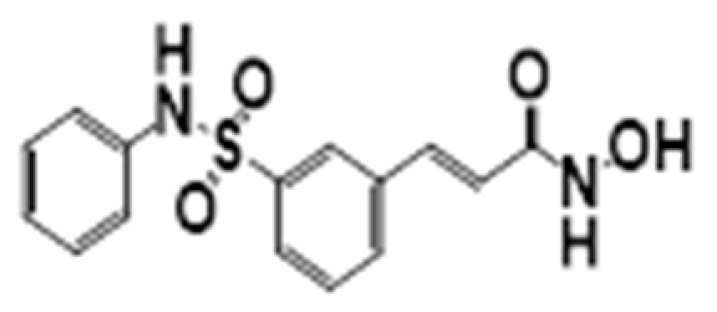	Thymic epithelial tumor and ovarian cancer	[31]
	Trichostatin	HDAC1-11	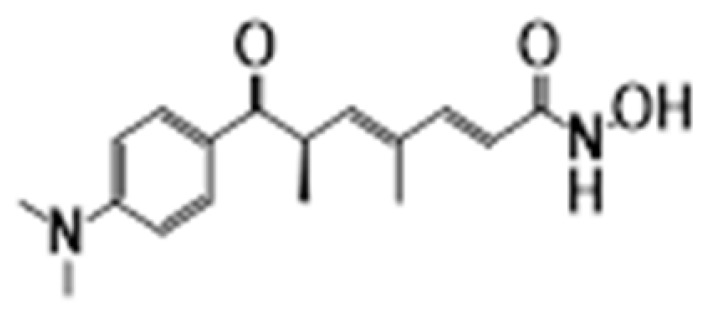	Colon and breast	[32,33]
Aliphatic acids	Pivanex	HDAC1-9	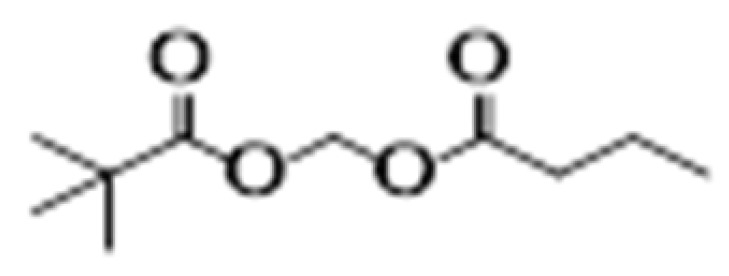	Liver carcinoma, Lung carcinoma and melanoma	[34]
	Valproic acid	HDAC1-9	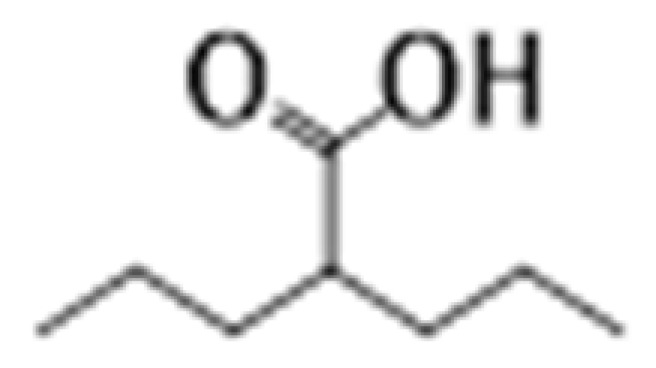	Prostate, breast and melanoma	[35]
Benzamides	Entinostat	HDAC1-3	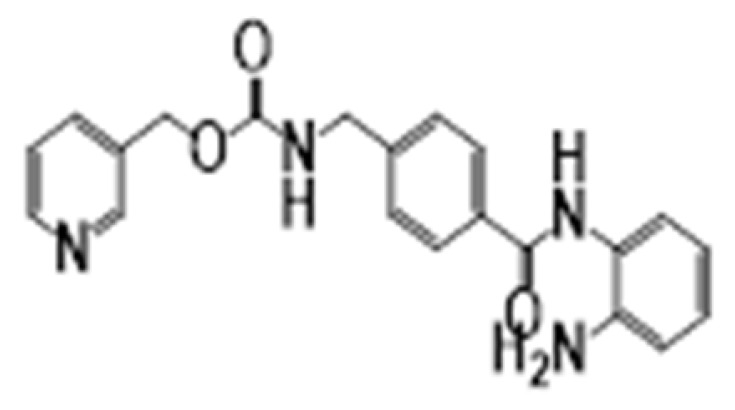	Breast cancer and advanced solid tumors	[36]
Electrophilic ketone	Trifluromethyl Ketones	HDAC4-9	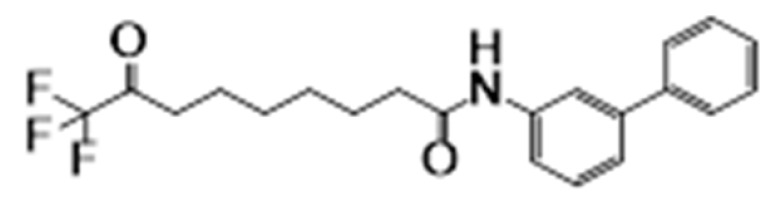	Prostate cancer	[37]
Cyclic tetrapeptides	Romidepsin	HDAC1-2	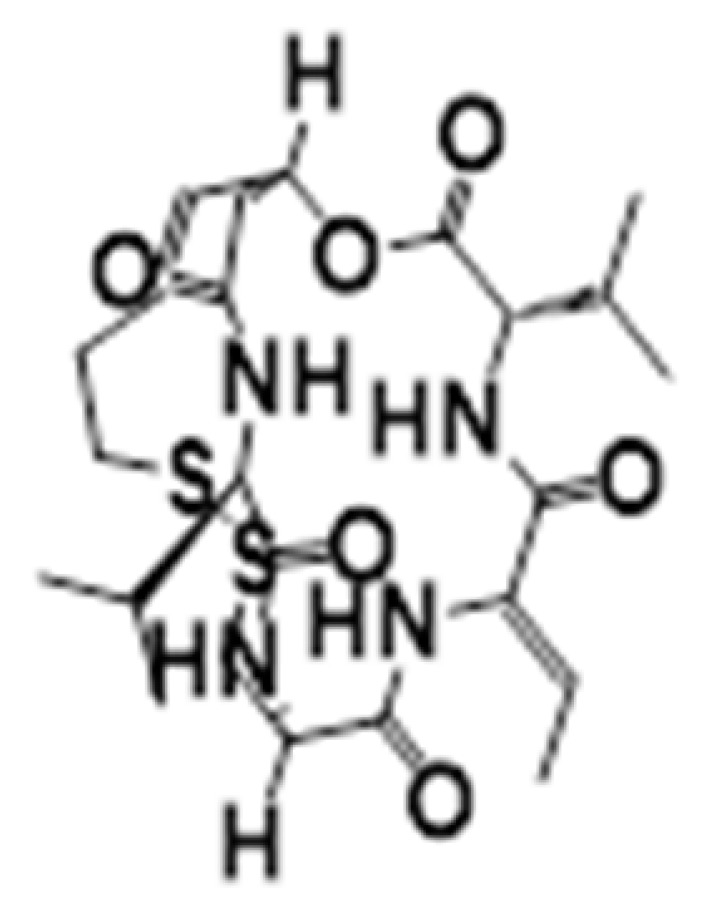	Thyroid, ovarian, pancreatic and breast cancer	[38]
Sirtuin inhibitors	Cambinol	SIRT1	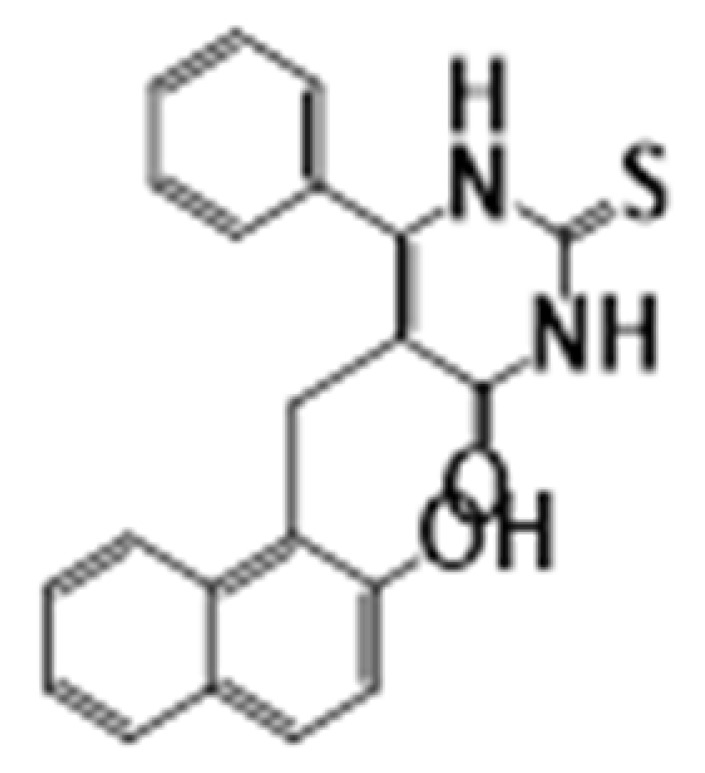	Sarcomas and lymphomas	[39]

Abbreviations: HDAC, histone deacetylase; SAHA, suberoylanilide hydroxamic acid; SIRT, sirtuin.

**Table 3 cancers-15-02808-t003:** Various HDAC inhibitors as drugs used for the treatment of AML and their mechanisms of action.

Classifications	Drug Name	Role in Treatment of AML	References
Hydroximates	Trichostatin A	Inhibits the pathway for DNA repair (NHEJ). Acetylation of repair factors and trapping of PARP1 at DNA double-strand brakes in chromatin, inducing leukemic toxicity. Possible synergistic effect of TSA with an inhibitor of PARP1.	[32,33]
	Vorinostat	Apoptosis and inhibition of cell growth. Increases differentiation induced by retinoic acid in acute promyelocytic leukemia cells. Induction of double-strand breaks and oxidative DNA damage.	[28]
	Panobinostat	Promotes apoptosis and inhibition of cell growth.	[29,30]
	Belinostat	Promotes cell cycle arrest, inhibits cell proliferation, and induces apoptosis.	[31]
Benzamides	Entinostat	Induces growth arrest and apoptosis. Downregulates antiapoptotic molecules BCL-2 and MCL-1, increases p21, and induces acetylation of H3.	[36]
Cyclic peptides	Romidepsin	Promotes apoptosis of chemo-resistant malignant cells and reversed their gene expression profile.	[38]
	Apicidin	Induces selective changes in P21WAF1/Cip1 and gelsolin gene expression, which control cell cycle and cell morphology.	[38]
Aliphatic acids	Valproic acid	Induces differentiation and inhibits proliferation and apoptosis of AML cells. No clinical effect when used as a single-agent therapy for AML. Synergistic effects with ATRA, decitabine, gemtuzumab ozogamicin, curcumin, hydroxyurea, 6-mercaptopurine, dasatinib, bortezomib, cytarabine	[35]

Abbreviations: AML, acute myeloid leukemia; BCL-2, B-cell leukemia/lymphoma 2 protein; MCL-1, myeloid cell leukemia-1; NHEJ, nonhomologous end-joining; PARP1, poly (ADP-ribose) polymerase 1; TSA, trichostatin.

**Table 4 cancers-15-02808-t004:** HDAC inhibitors of hydroxamate and non-hydroxamate families in different clinical trials.

Hydroxamate Family			Non-Hydroxamate Family		
Drug Name	Investigated Malignancy	Clinical Trial Phase	Drug Name	Investigated Malignancy	Clinical Trial Phase
Abexinostat	Skin cancers, non-small lung cancer, mantle cell lymphoma, acute myeloid leukemia	Phase-3	Tacedinaline	Solid and hematological cancers	Phase 3
Fimepinostat	lymphomas, brain tumors	Phase1/2	Entinostat	Gynecological cancers, CNS tumors, pancreatic cancer, non-small lung cancer	Phase 2
Quisinostat	JNJ26481585 ovarian cancer, non-small lung cancer	Phase 2	Domatinostat	Cutaneous T-cell lymphoma	Phase1
Ricolinostat	ACY-1215 lymphomas, Breast cancer, gynecological cancers	Phase 2	Givinostat	Polycythemia vera	Phase 2
Trichostatin A	Hematological cancers	Phase 1	KA2507	Melanoma	Phase 1
Nanatinostat	VRx-3996, EBV-associated cancers	Phase 1	Mocetinostat	Leiomyosarcoma and melanoma	Phase 2
CG200745	Myelodysplastic syndrome, pancreatic cancer	Phase1/2	OBP-801	Lung cancer	Phase 1a
Pracinostat	Prostate cancer, sarcoma, myelofibrosis, myelodysplastic syndrome	Phase 3	AR-42	Sarcoma and meningioma	Phase1
Resminostat	Pancreatic cancer, non-small lung cancer, colorectal carcinoma, Hodgkin’s lymphoma	Phase 2	Pivanex	Melanoma and lymphoblastic leukemia	Phase 2
CUDC-101	Advanced solid tumors	Phase 1	Givinostat	Polycythemia vera	Phase 2
MPT0E028	Advanced solid tumors	Phase 1			

Abbreviations: CNS, central nervous system; EBV, Epstein-Barr virus.

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
