# Peer review of "Potential of Synthetic and Natural Compounds as Novel Histone Deacetylase Inhibitors for the Treatment of Hematological Malignancies"

_cancers, 2023, doi:10.3390/cancers15102808_

Round 1
Reviewer 1 Report
The article you provided is well-written and heavy on the informative. It provides a good overview of the potential of histone deacetylase inhibitors (HDACi) to treat hematological malignancies. The authors do a good job of summarizing the current state of research in this area, and they identify several promising areas for future research. One of the strengths of the article is that it provides a balanced perspective on the potential of HDACi. The authors acknowledge that HDACi have the potential to be effective cancer therapies, but they also point out that there are some challenges that need to be addressed before HDACi can be widely used in clinical practice. For example, HDACi can have significant side effects, and they may not be effective against all types of cancer. Here is one criticism of the article:- The authors could have provided more information about the side effects of HDACi. and how natural compounds are better
good
Author Response
The authors of this manuscript express their sincere thanks to the reviewer for the critical assessment of this work. The authors have acted upon the recommendations of the reviewer which have resulted in a significant enhancement in the quality of this manuscript. All modifications incorporated in the manuscript are highlighted in a red color font. A “point-by-point” response to each comment is outlined below.
Comment 1:
The article you provided is well-written and heavy on the informative. It provides a good overview of the potential of histone deacetylase inhibitors (HDACi) to treat hematological malignancies. The authors do a good job of summarizing the current state of research in this area, and they identify several promising areas for future research. One of the strengths of the article is that it provides a balanced perspective on the potential of HDACi. The authors acknowledge that HDACi have the potential to be effective cancer therapies, but they also point out that there are some challenges that need to be addressed before HDACi can be widely used in clinical practice. For example, HDACi can have significant side effects, and they may not be effective against all types of cancer.
Response:
We greatly appreciate your role in reviewing this manuscript. Every attempt was made to provide a complete and unbiased review of the literature. Thank you for your comment.
Comment 2:
Here is one criticism of the article:
The authors could have provided more information about the side effects of HDACi. and how natural compounds are better
Response:
Additional information has been added regarding the side effects of HDAC inhibitors, along with further support for where natural compounds may be better options (page 2, lines 96-123).
Comment 3:
Despite the criticism, I think this is a well-written and informative article that provides a good overview of the potential of HDACi to treat hematological malignancies. I would recommend this article to anyone who is interested in learning more about this topic.
Response:
Thank you for your endorsement.
Additionally,
- The entire manuscript has been thoroughly checked and edited to minimize typographical errors as well as to ensure uniform style, organization, and quality.
- The reference list has been modified and many references are renumbered. Special attention is given to conform to the order of references and bibliographic style of the journal.
Finally,
On behalf of my co-authors, I once again express my sincere thanks to the erudite Managing Editor and reviewers for the valuable suggestions and constructive input to improve the quality of our manuscript.
Reviewer 2 Report
This review paper by Pal et al. summarized the histone deacetylase inhibitors for hematological Malignancies. This paper covered the some of the recent studies and was well organized.
The following points need to be addressed:
1. Provide references for the following claims:
Line 59: “Lysine acetylation was originally identified in the early 1960s when scientists noticed to large amount of PTMs in histones.”
Line 65: “These include chaperone proteins, transcription factors, hormone receptors, signaling mediators, and other proteins implicated in the DNA damage response.”
Line 162: “Other studies have shown evidence that HDACis may generate significant DNA damage, including DSBs, by increasing reactive oxygen species and oxidative stress and/or variant histone H2AX phosphorylation of serine, gH2AX, which is associated with DSBs.”
Line 165: “New studies have also demonstrated that modifications to chromatin may directly or indirectly result in DNA damage. This suggests the mechanisms responsible for chromatin remodeling and DNA damage signaling are intrinsically related.”
Section 5. Downregulation of DSB repair: “The implications of HDACis are greater than that of histone modification alone. Vorinostat (or SAHA) has been demonstrated to upregulate and maintain gH2AX expression in prostate and lung cancer cells by reducing the expression of MRE11 and HRR factor RAD50. In addition to RAD50, it was described to reduce the NHEJ proteins Ku70 and 258 Ku80 in melanoma, while NaB has been shown to downregulate Ku80, Ku70, and DNA PK. Additional genes downregulated in AML cells by HDACis MS275 and LAQ824 include RAD50, BRCA1, CHK2, EXO1, and Ku80.”
Line 340 : “In a study utilizing AML cells, researchers observed the effects of the new HDACis panobinostat (LBH589) in conjunction with doxorubicin.”
Line 434: “Additionally, MM cells pretreated with bortezomib were more susceptibleto the apoptosis and mitochondrial dysfunction caused by vorinostat. Indeed, this combination was effective against MM cells resistant to both dexamethasone and doxorubicin.”
Line 520: “Berberine was also demonstrated to hinder cancer spread by inhibiting transferase activity.”
Line 526: “the Connectivity Map (CMap) database and a gene expression signature-based method. It is hypothesized that berberine might inhibit protein synthesis, HDACs, and Akt/mTOR pathways, which was highly correlated with in silico predictions (Table 5).”
Line 593: “Specifically, ginsenosides displayed anticancer, anti-inflammatory, antioxidative, and vasorelaxant properties.”
Line 734: “Researchers purport xestoquinone's lethal action on Molt-4 cells to be caused by activation of many distinct apoptotic pathways such as death-receptor path way. Pretreatment of these cells with N-acetyl cysteine alleviated the loss of mitochondrial membrane potential, decreased apoptosis, and maintained topoisomerase I and II expression.”
Line 748: “Researchers also note an increase in cytochrome c release, caspase-3 activation, and the proteolytic cleavage of PARP. An uptick in A. camphorata-induced apoptosis coincided with a decrease in Bcl-2, a powerful inhibitor of cell death, and upregulation of Bax.”
2. Some references are not accurate.
Line 62: reference 5.
Table 2: Add the original publication of each inhibitor.
Line 178: Reference 25: It’s a paper about PROTAC design.
Line 229: Reference 44: The paper didn’t really talk about HRR and NHEJ.
Line 271: It’s not the paper that talked about the ChIP experiments.
Line 281: Reference 48 only talked about the HDAC expression level in human cancer tissue.
Line 391 and 397: Reference 64 and 65: These two studies didn’t use vorinostat.
Line 476: Reference 75 talked about using CXCR4 as a marker for AML.
Line 589: Reference 93 only talked about Xestoquinone.
Line 601: Reference 95 was not the research paper talking about 20(s)-Rh2.
Line 723: Reference 109 was a PK/PD study.
3. Provide more context in the figure legends to better explain the figures.
Minor points:
Table 2: Panobinostat instead of Panabinostat
Line 481: Furthermore instead of Fruthermore.
Line 497: NCT number of acute lymphoblastic leukemia and lymphoma
Line 691: BCL2L11 instead of BCL2,L11
Line 694: BNIP3L instead of BNIPL
Author Response
The authors of this manuscript express their sincere thanks to the reviewer for the critical assessment of this work. The authors have acted upon the recommendations of the reviewer which have resulted in a significant enhancement in the quality of this manuscript. All modifications incorporated in the manuscript are highlighted in a red color font. A “point-by-point” response to each comment is outlined below.
General comments:
This review paper by Pal et al. summarized the histone deacetylase inhibitors for hematological Malignancies. This paper covered the some of the recent studies and was well organized.
Response:
Thank you for your time and effort invested in reviewing this work. We have made changes based upon your valuable input.
Specific comments:
Major points:
Comment 1:
Provide references for the following claims:
- Line 59: “Lysine acetylation was originally identified in the early 1960s when scientists noticed to large amount of PTMs in histones.”
- Line 65: “These include chaperone proteins, transcription factors, hormone receptors, signaling mediators, and other proteins implicated in the DNA damage response.”
- Line 162: “Other studies have shown evidence that HDACis may generate significant DNA damage, including DSBs, by increasing reactive oxygen species and oxidative stress and/or variant histone H2AX phosphorylation of serine, gH2AX, which is associated with DSBs.”
- Line 165: “New studies have also demonstrated that modifications to chromatin may directly or indirectly result in DNA damage. This suggests the mechanisms responsible for chromatin remodeling and DNA damage signaling are intrinsically related.”
- Section 5. Downregulation of DSB repair: “The implications of HDACis are greater than that of histone modification alone. Vorinostat (or SAHA) has been demonstrated to upregulate and maintain gH2AX expression in prostate and lung cancer cells by reducing the expression of MRE11 and HRR factor RAD50. In addition to RAD50, it was described to reduce the NHEJ proteins Ku70 and 258 Ku80 in melanoma, while NaB has been shown to downregulate Ku80, Ku70, and DNA PK. Additional genes downregulated in AML cells by HDACis MS275 and LAQ824 include RAD50, BRCA1, CHK2, EXO1, and Ku80.”
- Line 340 : “In a study utilizing AML cells, researchers observed the effects of the new HDACis panobinostat (LBH589) in conjunction with doxorubicin.”
- Line 434: “Additionally, MM cells pretreated with bortezomib were more susceptibleto the apoptosis and mitochondrial dysfunction caused by vorinostat. Indeed, this combination was effective against MM cells resistant to both dexamethasone and doxorubicin.”
- Line 520: “Berberine was also demonstrated to hinder cancer spread by inhibiting transferase activity.”
- Line 526: “the Connectivity Map (CMap) database and a gene expression signature-based method. It is hypothesized that berberine might inhibit protein synthesis, HDACs, and Akt/mTOR pathways, which was highly correlated with in silico predictions (Table 5).”
- Line 593: “Specifically, ginsenosides displayed anticancer, anti-inflammatory, antioxidative, and vasorelaxant properties.”
- Line 734: “Researchers purport xestoquinone's lethal action on Molt-4 cells to be caused by activation of many distinct apoptotic pathways such as death-receptor path way. Pretreatment of these cells with N-acetyl cysteine alleviated the loss of mitochondrial membrane potential, decreased apoptosis, and maintained topoisomerase I and II expression.”
- Line 748: “Researchers also note an increase in cytochrome c release, caspase-3 activation, and the proteolytic cleavage of PARP. An uptick in A. camphorata-induced apoptosis coincided with a decrease in Bcl-2, a powerful inhibitor of cell death, and upregulation of Bax.”
Response:
We sincerely apologize for missing the references. They have been updated as follows:
- Ref number 5 (page 2, lines 59-60; page 28, lines 868-869).
- Ref number 6 (page 2, lines 65-67; page 28, lines 870).
- Ref number 41 (page 7, lines 189-192; page 30, lines 947-949).
- Ref number 42 (page 7, lines 192-195; page 30, lines 950-951).
- Ref number 67 (page 11, lines 284-28; page 31, lines 1007-1009).
- Ref number 78 (page 14, lines 371-372; page 31, lines 1032-1034).
- Ref number 93 (page 16, lines 465-467; page 32, lines 1072-1073).
- Ref number 102 (page 20, lines 551-553; page 32, lines 1093-1094).
- Ref number 103 (page 20, lines 557-560; page 32, lines 1095-1097).
- Ref number 117 (page 23, lines 625-627; page 33, lines 1131-1132).
- Ref number 115 (page 26, lines 765-771; page 33, lines 1125-1127).
- Ref number 135 (page 26, lines 780-783; page 34, lines 1174-1175).
Comment 2:
Some references are not accurate.
- Line 62: reference 5.
- Table 2: Add the original publication of each inhibitor.
- Line 178: Reference 25: It’s a paper about PROTAC design.
- Line 229: Reference 44: The paper didn’t really talk about HRR and NHEJ.
- Line 271: It’s not the paper that talked about the ChIP experiments.
- Line 281: Reference 48 only talked about the HDAC expression level in human cancer tissue.
- Line 391 and 397: Reference 64 and 65: These two studies didn’t use vorinostat.
- Line 476: Reference 75 talked about using CXCR4 as a marker for AML.
- Line 589: Reference 93 only talked about Xestoquinone.
- Line 601: Reference 95 was not the research paper talking about 20(s)-Rh2.
- Line 723: Reference 109 was a PK/PD study.
Response:
Thank you for your comment. The references have been updated in the text and references list. We fixed the errors with the references as follows:
- Ref number 5 has been corrected in the reference list (page 28, lines 868-869).
- Ref number 28-39 (page 6, lines 181-183; page 29, lines 918-944).
- Ref number 44 (page 7, line 205; page 30, lines 954-955).
- Ref number 64 (page 10, line 258; page 31, lines 1001-1002).
- Ref number 68 (page 11, line 300; page 31, lines 1010-1012).
- Ref number 69 (page 11, line 310; page 31, line 1013).
- Ref number 86, 87 (page 15, line 422 and lines 424-428; page 32, lines 1053-1057).
- Ref number 97 (page 16, line 489; page 32, lines 1082-1084).
- Ref number 114 (page 23, line 620; page 33, lines 1122-1124).
- Ref number 118 (page 23, line 633; page 33, lines 1133-1134).
- Ref number 132 (page 26, line 755; page 34, lines 1166-1168).
Comment 3:
Provide more context in the figure legends to better explain the figures.
Response:
Necessary corrections have been made in the revised manuscript (page 12, lines 325-326; page 14, lines 368-370).
Minor points:
Comment 1:
Table 2: Panobinostat instead of Panabinostat
Response:
We have corrected the spelling mistake in the revised manuscript (page 6).
Comment 2:
Line 481: Furthermore instead of Fruthermore.
Response:
We have corrected the spelling mistake (page 17, line 513).
Comment 3:
Line 497: NCT number of acute lymphoblastic leukemia and lymphoma
Response: Correct NCT number of acute lymphoblastic leukemia and lymphoma is mentioned (page 18, line 529).
Comment 4:
Line 691: BCL2L11 instead of BCL2,L11
Response:
We have made the necessary corrections (page 25, lines 723 and 725).
Comment 5:
Line 694: BNIP3L instead of BNIPL (line 722)
Response:
We have made the corrections in the revised manuscript (page 25, lines 725-726).
Additionally,
- The entire manuscript has been thoroughly checked and edited to minimize typographical errors as well as to ensure uniform style, organization, and quality.
- The reference list has been modified and many references are renumbered. Special attention is given to conform to the order of references and bibliographic style of the journal.
Finally,
On behalf of my co-authors, I once again express my sincere thanks to the erudite Managing Editor and reviewers for the valuable suggestions and constructive input to improve the quality of our manuscript.
Reviewer 3 Report
Please find below my comments for the review.
Natural Compounds as Novel Histone Deacetylase Inhibitors 2 for the Treatment of Hematological Malignancies
The manuscript is well organized and highlights the importance of different natural products-based HDACis as therapeutic agents in the treatment on hematological malignancies.
The manuscript is written in correct and understandable English, and no language revision is requested.
The references are quite updated even if it would be appropriate to insert some more recent ones (in the 2018-2023 period).
I don’t have revisions as a request, the work is well done and organized, as well as very interesting.
There are no other comments to add.
Best regards

Author Response
The authors of this manuscript express their sincere thanks to the reviewer for the critical assessment of this work. The authors have acted upon the recommendations of the reviewer which have resulted in a significant enhancement in the quality of this manuscript. All modifications incorporated in the manuscript are highlighted in a red color font. A “point-by-point” response to each comment is outlined below.
Comment 1:
The manuscript is well organized and highlights the importance of different natural products-based HDACis as therapeutic agents in the treatment on hematological malignancies.
Response:
Thank you for your feedback. We are appreciative of the effort you invested in reviewing this work.
Comment 2:
The manuscript is written in correct and understandable English, and no language revision is requested.
Response:
We thank the reviewer for checking our manuscript.
Comment 3:
The references are quite updated even if it would be appropriate to insert some more recent ones (in the 2018-2023 period).
Response:
We have made attempts to include additional updated references wherever possible.
Comment 4:
I don’t have revisions as a request, the work is well done and organized, as well as very interesting.
Response:
We appreciate this generous comment.
Comment 5:
There are no other comments to add.
Response:
Thank you for your time and effort.
Additionally,
- The entire manuscript has been thoroughly checked and edited to minimize typographical errors as well as to ensure uniform style, organization, and quality.
- The reference list has been modified and many references are renumbered. Special attention is given to conform to the order of references and bibliographic style of the journal.
Finally,
On behalf of my co-authors, I once again express my sincere thanks to the erudite Managing Editor and reviewers for the valuable suggestions and constructive input to improve the quality of our manuscript.
Round 2
Reviewer 2 Report
Thanks for addressing all the comments. The authors provide a good overviews of the HDAC inhibitors for hematological malignancies. The article is well-organized and informative.